# C. elegans GLP-1/Notch activates transcription in a probability gradient across the germline stem cell pool

ChangHwan Lee[1,2], Erika B Sorensen[1,2†], Tina R Lynch[2], Judith Kimble[1,2]*

[1]Howard Hughes Medical Institute, University of Wisconsin-Madison, Madison, United States; [2]Department of Biochemistry, University of Wisconsin-Madison, Madison, United States

**Abstract** *C. elegans* Notch signaling maintains a pool of germline stem cells within their single-celled mesenchymal niche. Here we investigate the Notch transcriptional response in germline stem cells using single-molecule fluorescence *in situ* hybridization coupled with automated, high-throughput quantitation. This approach allows us to distinguish Notch-dependent nascent transcripts in the nucleus from mature mRNAs in the cytoplasm. We find that Notch-dependent active transcription sites occur in a probabilistic fashion and, unexpectedly, do so in a steep gradient across the stem cell pool. Yet these graded nuclear sites create a nearly uniform field of mRNAs that extends beyond the region of transcriptional activation. Therefore, active transcription sites provide a precise view of where the Notch-dependent transcriptional complex is productively engaged. Our findings offer a new window into the Notch transcriptional response and demonstrate the importance of assaying nascent transcripts at active transcription sites as a readout for canonical signaling.

*For correspondence: jekimble@wisc.edu

Present address: †Department of Biology, Wabash College, Crawfordsville, United States

Competing interests: The authors declare that no competing interests exist.

## Introduction

Cell-cell signaling lies at the heart of animal development and homeostasis, while its dysfunction can lead to disease (e.g. *Housden and Perrimon, 2014*). Most canonical cell signaling pathways (e.g. Notch, Wnt, Hedgehog, TGF-β) regulate transcription and major advances have been made unraveling their molecular mechanisms. Yet virtually no studies of any canonical signaling pathway have tackled the question of how they modulate nuclear transcription (i.e. generation of nascent transcripts at active transcription sites) during animal development. Does signaling orchestrate a synchronous nuclear response? Or does it tune the probability of transcriptional firing and number of nascent transcripts produced? Answers to these questions are fundamental to our molecular understanding of intercellular signaling.

To analyze the effect of intercellular signaling in its natural context during development, we focus on Notch signaling and its regulation of a stem cell pool in the nematode *C. elegans*. The Notch pathway is strongly conserved across animal phylogeny (e.g. *Gazave et al. 2009*). Moreover, like other canonical signaling pathways, Notch is central to development and homeostasis, with dysfunction or misregulation causing disease (e.g. cancer, stroke) (*Bray, 2006*; *Roy et al., 2007*; *Kopan and Ilagan, 2009*; *Tien et al., 2009*; *de la Pompa and Epstein, 2012*). The molecular mechanism of Notch signaling is essentially the same across metazoans, including nematodes, flies and vertebrates. Briefly, Notch ligands trigger cleavage of the Notch receptor to release its intracellular domain (NICD), which assembles into a nuclear complex to activate transcription (*Kopan and Ilagan, 2009*). The transcriptional response to Notch signaling has been analyzed in diverse systems (e.g. *Shimojo et al., 2008*; *Ilagan et al. 2011*; *Hoyle and Ish-Horowicz, 2013*; *Imayoshi et al., 2013*;

*Kershner et al., 2014*; *Jenkins et al., 2015*), but no analyses have been done at the resolution of nascent transcripts at active transcription sites in the nucleus. Transcriptional reporters are a common readout of the Notch transcriptional response (*Hansson et al., 2006*; *Imayoshi et al., 2013*), but they are both artificial and indirect. Conventional RNA *in situ* hybridization to endogenous Notch targets has also been used as a readout of Notch signaling, for example during segmentation of the zebrafish embryo (*Hoyle and Ish-Horowicz, 2013*), but this method cannot resolve individual chromosomal loci and typically cannot resolve individual cells.

Our focus on Notch signaling in the *C. elegans* gonad was inspired largely by unanswered questions about the spatial extent of Notch activity in this system. It had been known for some time that Notch, known as GLP-1/Notch in this system, is crucial for the regulation of the germline stem cell pool and establishing polarity in the germline (*Austin and Kimble, 1987*; *Kimble and Crittenden, 2007*). The single-celled mesenchymal niche, called the distal tip cell (DTC), uses Notch signaling to maintain a pool of germline stem cells (GSCs) at the distal end of the 'progenitor zone' in the distal gonad (*Figure 1A,B*) (*Kimble and White 1981*; *Crittenden et al., 2006*; *Cinquin et al., 2010*; *Byrd et al., 2014*). Previous studies suggested that Notch signaling might function throughout the GSC pool or even beyond. Transcripts from key Notch target genes are expressed in the distal gonad (*Kershner et al., 2014*), but this previous study did not define expression at the level of single cells, much less at single chromosomal loci. The site of Notch signaling is ambiguous, because of the elaborate architecture of the signaling cell — the DTC cell body caps the distal gonad and extends ultra-thin processes intercalating throughout the GSC pool as well as long external processes along the entire progenitor zone and sometimes beyond (*Fitzgerald and Greenwald, 1995*; *Hall et al., 1999*; *Crittenden et al., 2006*; *Byrd et al., 2014*). Which parts of the DTC are actually signaling and how sustained is the transcriptional response once the receptor is cleaved? Answers to these questions are critical to understanding how GLP-1/Notch regulates this stem cell pool and will be of heuristic value for other systems.

Our decision to focus on the *C. elegans* gonad was also based on its unique tractability for analyzing active transcription sites responsive to canonical intercellular signaling. Visualization of nascent transcripts has been technically possible for some time, due to the advent of single-molecule fluorescence *in situ* hybridization (smFISH) (*Raj et al., 2008*), but the application of this method to intercellular signaling has faced numerous challenges. Signaling and receiving cells within an intact organism are often poorly defined; signaling is often transient or subject to changing regulatory circuitry; and signaling in tissue culture cells typically relies on overexpressed components and reporter readouts, either of which has potential to alter or mask the response. Several features of Notch signaling in the *C. elegans* gonad circumvent these issues. Not only is the signaling cell defined, but signaling is continuous rather than transient. Thus, the DTC expresses DSL ligands in both larvae and adults (*Henderson et al., 1994*; *Nadarajan et al., 2009*); mitotically dividing germ cells express the GLP-1/Notch receptor at all stages (*Crittenden et al., 1994*), and GLP-1/Notch signaling controls GSC maintenance throughout life of the animal (*Austin and Kimble, 1987*; *Morgan et al., 2010*). In addition, two direct Notch target genes are available for assays: *sygl-1* (*synthetic Glp*) and *lst-1* (*lateral signaling target)* encode key stem cell regulators, whose removal mimics the loss of Notch signaling (*Kershner et al., 2014*) (*Figure 1C*). Finally, GLP-1/Notch appears to regulate a pool of cells, enabling quantitation of many receiving cells at one time. This feature would be particularly useful if Notch affects probability of transcriptional firing, as might be expected. Together, these features circumvent common challenges and establish the *C. elegans* gonad as a superb model for visualizing Notch-dependent transcriptional activation in the nucleus.

Here we use smFISH and a custom MATLAB code to visualize and quantitate the transcriptional response to Notch signaling in the *C. elegans* distal gonad. We have been able to distinguish nascent transcripts at active transcription sites (ATS) in the nucleus as well as mature mRNAs in the cytoplasm, both from endogenous target genes in wild-type gonads. Therefore, this study reveals the response to Notch signaling in its natural context – no reporters and no manipulated signaling components. We find that Notch regulates both the probability of transcriptional firing at individual chromosomal loci and the number of nascent transcripts produced at active transcription sites. Surprisingly, we discover a gradient in the probability of generating a Notch-regulated ATS across the GSC pool. Yet the steeply graded ATS create a more uniform field of mRNAs that extends beyond the gradient. Therefore, nuclear ATS provide a direct measure for effective engagement of the Notch-dependent transcription complex at its target genes. Finally, we analyze the transcriptional

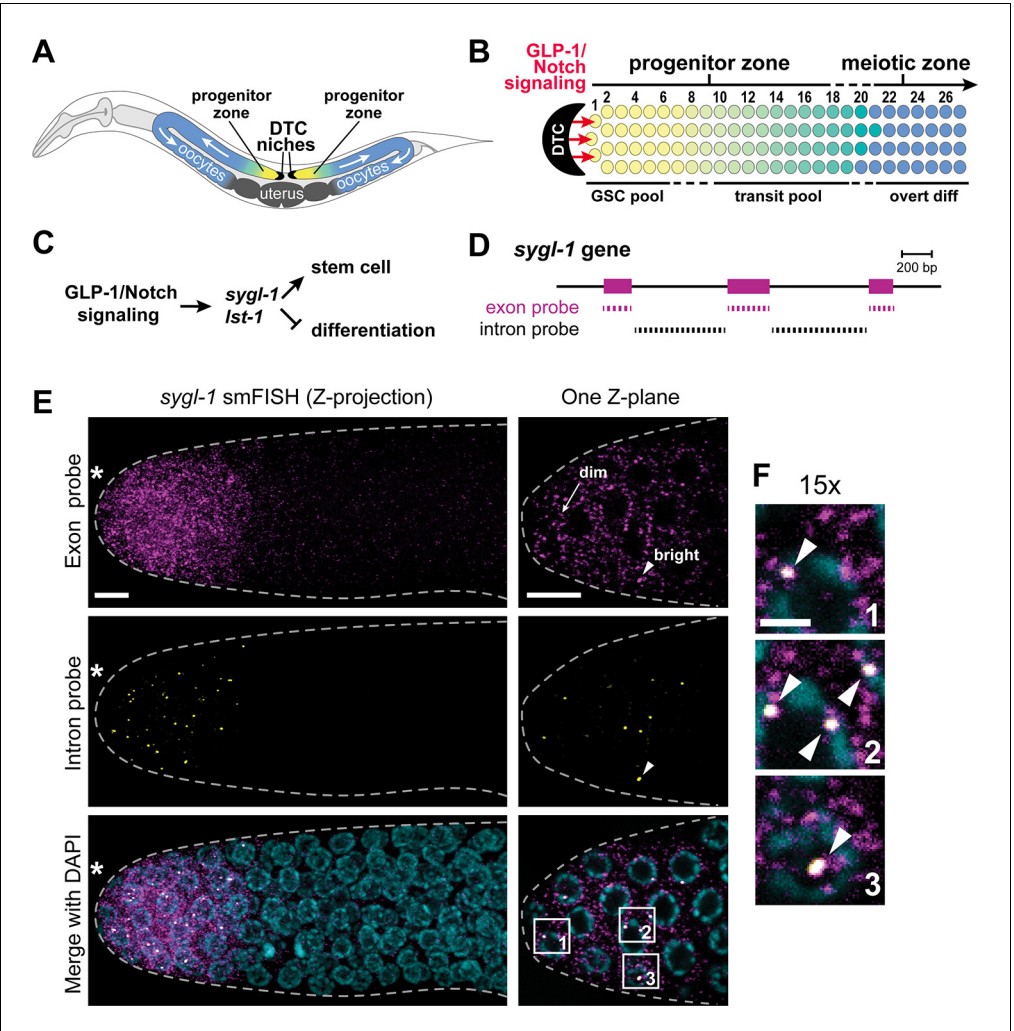

**Figure 1.** Visualization of *sygl-1* transcripts using smFISH. (**A**) Schematic of adult *C. elegans* hermaphrodite with two U-shaped gonadal arms, each with a single-celled niche (DTC, black crescent) and a progenitor zone of mitotically dividing germ cells at the distal end. Germ cell movement is from distal to proximal (white arrows). Somatic gonadal structures are located centrally (dark grey). (**B**) Organization of germ cells in distal gonad. The only somatic cell in the distal gonad is the DTC; diagrammed here is its cell body (see Introduction for more about DTC architecture). The progenitor zone includes a distal pool of naïve undifferentiated germ cells (yellow), which have been proposed to constitute the GSC pool, and more proximal germ cells (yellow to green transition), which have been triggered to differentiate and are maturing as they transit towards overt differentiation (*Cinquin et al., 2010*). Transit germ cells divide only once or twice before entering the meiotic cell cycle (*Fox and Schedl, 2015*). The boundary between progenitor and meiotic zones is not sharp (dashed line), and similarly, the boundaries of GSC and transit pools are not sharp (dashed lines). Positions of germ cells are conventionally designated as the number of 'germ cell diameters' along the distal-proximal axis from the distal end, with position 1 being immediately adjacent to the DTC cell body; the transition from GSC to transit pools is proposed to occur at position 6–8 (*Cinquin et al., 2010*), and from progenitor to meiotic zone at position 19–22 (*Crittenden et al., 1994*). (**C**) The *sygl-1* and *lst-1* genes are direct targets of GLP-1/Notch signaling and key regulators of germline stem cell maintenance (*Kershner et al., 2014*). (**D**) Schematic of *sygl-1* exon/intron structure. Exon-specific (magenta) and intron-specific (black) probes for single-molecule RNA FISH (smFISH) were labeled with different fluors (see Materials and methods). (**E-F**) *sygl-1* smFISH in distal gonad. Exon probes (magenta); intron probes (yellow). DAPI marks nuclei (blue). Nuclei have DAPI-free centers because of their large nucleoli. Merge (bottom) is an overlay of exon probe, intron probe and DAPI channels. *Figure 1—figure supplement 1A* shows *sygl-1* smFISH in a whole gonad. (**E**) Distal gonad dissected from wild-type adult (24 hr post mid-L4 stage), showing dim spots in the cytoplasm (arrow) and bright spots in the nucleus (arrowhead). Grey dashed line marks gonadal outline; asterisk marks distal end of gonad; scale bar is 5 µm. (**F**) 15X magnification of nuclei within boxes in *Figure 1E*, bottom panel. The merged images show overlap of exon and intron signals as white spots

*Figure 1 continued on next page*

*Figure 1 continued*

(arrowheads), which not only overlap with DAPI, as shown here, but are also within nuclei as assayed using a nuclear lamin (*Figure 1—figure supplement 1B*). Scale bar: 2 μm. The specificity of smFISH probes is confirmed in *Figure 1—figure supplement 1C,D* and Notch dependence of smFISH signals in *Figure 1—figure supplement 2*.

The following figure supplements are available for figure 1:

**Figure supplement 1.** *sygl-1* smFISH visualization of nascent nuclear transcripts and mature cytoplasmic transcripts.

**Figure supplement 2.** GLP-1/Notch-dependence of *sygl-1* ATS.

response to two well-characterized Notch mutants. A weaker than wild-type NICD lowers the extent of the Notch-dependent transcriptional response, including the magnitude of transcription probability, number of transcripts generated per active transcription site and range of the spatial gradient, a finding with implications for both patterning and disease. By contrast, an unregulated Notch receptor drives an ectopic transcriptional response at an essentially wild-type level of intensity.

## Results

### Visualization of *sygl-1* active transcription sites (ATS) and cytoplasmic mRNAs

To visualize the GLP-1/Notch transcriptional response in wild-type gonads, we monitored *sygl-1* transcripts with two probe sets, designed to either *sygl-1* exons or introns and coupled to spectrally-distinct fluorophores (*Figure 1D*). As reported previously using conventional *in situ* hybridization (*Kershner et al., 2014*), we detected *sygl-1* RNAs in both the distal and proximal gonad (*Figure 1—figure supplement 1A*); here we focus on the distal *sygl-1* RNAs, because only they are Notch-dependent. The *sygl-1* exon probe set revealed a multitude of spots in germ cells within the niche, with most spots dim but a few bright (*Figure 1E*, top); by contrast, the intron probe detected many fewer spots that were mostly bright (*Figure 1E*, middle). Importantly, foci seen with the intron probe overlapped with the brighter spots seen with the exon probe (*Figure 1F*); both also overlapped with DAPI and were therefore nuclear (*Figure 1E,F*). Nuclear localization was confirmed by labeling with both smFISH and an antibody to the nuclear lamina (*Figure 1—figure supplement 1B*). The same pattern of signals was observed either when the exon and intron probe sets were used together or independently.

The *sygl-1* exon probes likely highlight Notch-dependent nascent transcripts in the nucleus as well as mature mRNAs in the cytoplasm, while intron probes likely mark only nascent nuclear transcripts. Two experiments confirmed that interpretation. First, treatment with α-amanitin, a standard RNA polymerase II inhibitor, abolished the nuclear dots for both exon and intron probes, but not the cytoplasmic dots (*Figure 1—figure supplement 1C*). Therefore, nuclear dots are nascent transcripts and identify active transcription sites (ATS). Second, *sygl-1* RNAi eliminated the cytoplasmic but not the nuclear dots seen with exon probes (*Figure 1—figure supplement 1D*). Because RNAi depletes mature but not primary transcripts (*Maamar et al., 2013*), the cytoplasmic dots must be mRNAs; this experiment also confirms the specificity of the *sygl-1* exon probe. Finally, we confirmed Notch dependence: *sygl-1* smFISH signals were not found in *glp-1* null mutants (*Figure 1—figure supplement 2A,B*) whereas transcription from a non-target gene, *let-858*, was unaffected (*Figure 1—figure supplement 2C,D*); moreover, *sygl-1* smFISH signals expanded in a *glp-1* gain-of-function mutant (*Figure 1—figure supplement 2E,F*). We conclude that *sygl-1* smFISH provides a high-resolution view of Notch-activated nascent transcripts in the nucleus and Notch-dependent mature mRNAs in the cytoplasm.

## Quantitation of *sygl-1* ATS and cytoplasmic mRNAs

We developed a custom MATLAB code (*Source code 1*) to score the presence, intensity and spatial distribution of smFISH signals in confocal images of adult gonads (*Figure 2A*). Briefly, the code uses DIC to define the tissue edge and DAPI to generate a 3-D reconstruction of nuclei (*Figure 2A*). Positions were recorded in 3-D, but are reported here as sites along the gonadal distal-proximal axis in microns (μm) from the distal end; microns were then translated into the more conventional metric of 'germ cell diameters (gcd)' from the distal end (*Figure 1B*). The code excludes the DTC nucleus (*Figure 2A*) and includes all germ cell nuclei within the field of view, corresponding to roughly 60 μm or 13 gcd from the distal end (*Figure 1E*). This field of view includes all *sygl-1* ATS and mRNAs in the distal gonad. Following tissue and nuclear reconstruction, we used the code to quantify smFISH signals as a function of both subcellular location (cytoplasm versus nucleus) and position in the gonad (μm/gcd from distal end). As expected, spots seen with the exon probe fell into two groups: one with bright spots co-localizing with both DAPI and the intron-probe-detected ATS, and the other with dimmer spots found outside DAPI and failing to co-localize with intron-probe-detected ATS. Signal intensities of individual spots were normalized to background in each gonad, and then pooled from 78 gonads after normalization between gonads (see Materials and methods). The intensities of the *sygl-1* cytoplasmic mRNAs clustered tightly (*Figure 2B*), a nearly uniform signal that likely corresponds to a single mRNA (*Raj et al., 2008*; *Raj and Tyagi, 2010*; *Lee et al., 2013*; *Lee et al., 2015*; *Padovan-Merhar et al., 2015*; *Lee et al., 2016*). By contrast, *sygl-1* ATS intensities spanned a broad range using either the exon (*Figure 2C*) or intron (*Figure 2D*) probe. Comparison of nuclear and cytoplasmic signal intensities, obtained with the same probe set (exon), revealed a mean ATS intensity roughly 17-fold greater on average than the mean single mRNA intensity (*Figure 2B,C*). This 17-fold value suggests that roughly 17 nascent transcripts are generated at individual *sygl-1* ATS, on average. This estimate is rough because ATS intensities combine signals from short beginning transcripts and longer elongating transcripts. The progenitor zone includes ~4% germ cells in G1, ~71% in S-phase, ~22% in G2 and ~3% in M-phase (*Seidel and Kimble, 2015*). Indeed, the maximum number of ATS per nucleus was four (*Figure 2E*), as expected for the maximum number of *sygl-1* chromosomal loci in this population of actively dividing cells.

The presence of some nuclei with ATS and others without ATS suggested that Notch might activate transcription in bursts and do so independently at individual chromosomal loci, as seen in other systems (e.g. *Elowitz et al., 2002*; *Chubb et al., 2006*; *Raj et al., 2006*; *Boettiger and Levine, 2009*). Indeed, most nuclei with any ATS had only one, while some had three (*Figure 2E*), demonstrating that individual *sygl-1* loci within the same nucleus could be activated independently. We also compared ATS intensities within single nuclei in all pairwise combinations, relying on those nuclei with two or more ATS. The intensities of ATS pairs failed to correlate with each other (*Figure 2F*), consistent with independent gene activation within the nucleus of a receiving cell.

We wondered if the presence of an ATS might correlate with cell cycle stage, a possibility that might have affected why ATS occurred in a subset of nuclei. We first confirmed in our 3-D reconstructed gonads that nuclear size was a good measure of cell cycle progression (*Figure 2—figure supplement 1A*), as previously established (*Seidel and Kimble, 2015*). We also confirmed in the reconstructed gonads that cell cycle stage was not linked to position along the gonadal axis (*Figure 2—figure supplement 1B,C*), also as previously established (*Crittenden et al., 2006*). We then assessed cells with and without ATS for both DNA content and nuclear size, but found no significant difference for either cell cycle measure (*Figure 2G,H*). To ask if cell cycle stage might instead affect the full *sygl-1* output per nucleus, we summed ATS signal intensities in each nucleus, but found no correlation between those summed values and either DNA content or nuclear size (*Figure 2I,J*). We conclude that the sporadic appearance of Notch-dependent ATS cannot be explained by a link with cell cycle stage.

## Gradient of Notch-dependent *sygl-1* transcriptional activation

To learn where within the progenitor zone GLP-1/Notch is engaged in transcriptional activation, we analyzed *sygl-1* ATS as a function of position along the distal-proximal axis of the distal gonad (*Figure 3A–F*). The vast majority of *sygl-1* ATS were restricted to a region of 1–30 μm (~1–7 gcd) from the distal end (*Figure 3A–F*, dashed red line) and their distribution was graded within that region. Thus, the percentage of ATS-positive nuclei was graded from ~70% at the distal-most end

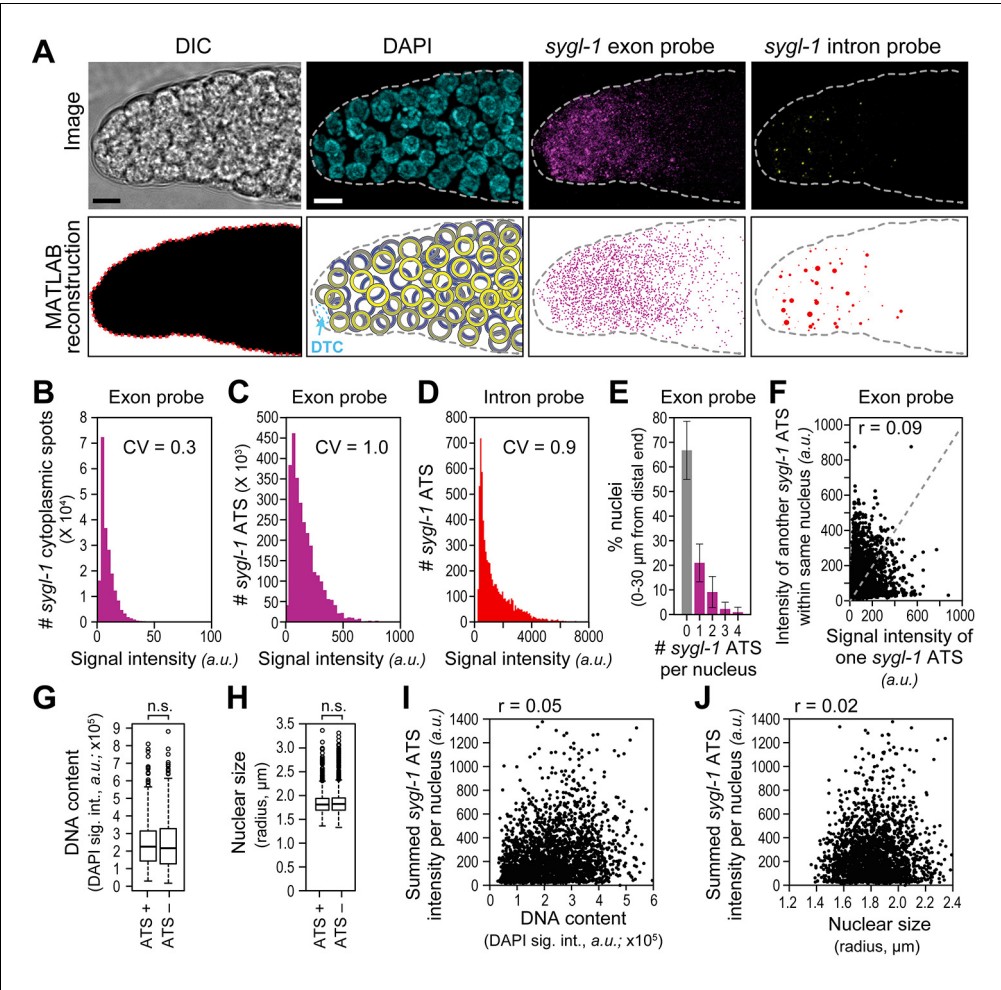

**Figure 2.** The *sygl-1* transcriptional response to Notch signaling. (**A**) MATLAB reconstruction of distal gonad. Top row, DIC image is one Z-plane; all others are Z-projections. Far left, DIC was used to determine gonadal outline; middle-left, DAPI reveals nuclei (blue); middle-right, signal from *sygl-1* exon probes shows *sygl-1* cytoplasmic mRNAs and nuclear active transcription sites (ATS) (magenta); far right, signal from *sygl-1* intron probes show only ATS (yellow). Bottom row, outputs from custom MATLAB code (***Source code 1***). Far left, gonadal outline (red dashed line), middle-left, germ cell nuclei false-colored in a depth gradient (yellow-blue) according to Z-position and with DTC (cyan) excluded; middle-right, cytoplasmic mRNAs (magenta) with nuclear ATS excluded computationally; far right, ATS (red) with smFISH signals scaled according to intensity. Scale bar: 5 μm. (**B**) Signal intensities of cytoplasmic dots from *sygl-1* exon probes. A total of 222,260 spots were analyzed from 78 gonads. Raw values from Z-planes were normalized to background levels in the same plane and the mean intensity value set to 10 arbitrary units (*a.u.*) for each gonad. CV: coefficient of variation (CV <1, significantly narrow distribution). (**C,D**) Signal intensities of *sygl-1* ATS. A total of 2627 spots were analyzed from 78 gonads. (**C**) Intensities of *sygl-1* ATS using exon probes. We first normalized raw values to background levels in the same Z-plane and then normalized to mean intensity of *sygl-1* cytoplasmic spots seen with the same probe in the same gonad. Mean intensity of *sygl-1* ATS was 172.0 *a.u.*, or roughly 17-fold more than the mean intensity of *sygl-1* individual mRNAs. (**D**) Intensities of *sygl-1* ATS using intron probes. Raw values were normalized to background levels in the same Z-plane. (**E**) Number of *sygl-1* ATS per nucleus in 7018 nuclei (78 gonads). Error bars: standard deviation. (**F**) Pair-wise comparisons of *sygl-1* ATS intensities within one nucleus (78 gonads), using normalized values from exon probe. Each black dot represents one pairing. Grey dashed line indicates a perfect correlation (Pearson's correlation coefficient r = 1); r indicates the correlation coefficient from data in the graph. (**G-J**) *sygl-1* transcriptional activity is independent of cell cycle stage. The cell cycle stage was monitored for DNA content (summed DAPI signal) or nuclear size, which are correlated (***Figure 2—figure supplement 1A***), in all nuclei located 0–30 μm (1–7 gcd) from the distal end (n = 6979 nuclei total); n.s.: not significant (p>0.05) by t-test. The cell cycle is also independent from nuclear location in the gonad (***Figure 2—figure supplement 1B,C***). (**G,H**) DNA content (**G**) or nuclear size (**H**) was compared between ATS-positive and ATS-negative cells. For all box-and-

*Figure 2 continued on next page*

*Figure 2 continued*

whisker plots in this study, the bold line in the box shows the median; top and bottom of box are the third and first quartiles, respectively; whiskers, maximum and minimum of data points; circles, outliers (value greater than 1.5X first or third quartile from the median). (I, J) Summed *sygl-1* ATS intensity was estimated by pooling all ATS signal intensities (*a.u.*) within the same nucleus. Each black dot represents a single nucleus.

The following figure supplement is available for figure 2:

**Figure supplement 1.** Cell cycle analysis in reconstructed gonads.

to <5% at the proximal edge of the region with most ATS (*Figure 3A*). ATS-negative nuclei were not likely a detection artifact, because *sygl-1* mRNAs were detectable in the cytoplasm of those same ATS-negative cells (see below). Therefore, the probability of ATS firing is graded within the distal third of the progenitor zone. Consistent with this probability gradient, we also saw a gradient when these same data were graphed as number of *sygl-1* ATS per nucleus (*Figure 3B*), percentages of nuclei with >1 ATS per nucleus (*Figure 3C*) and average ATS number (*Figure 3D*).

We also investigated intensities of individual *sygl-1* ATS as a function of position. As described above, ATS signal intensity provides a measure of the number of nascent transcripts, but only an estimate. Although ATS signal intensities varied considerably (*Figure 3E*), their mean was relatively constant across the distal third of the progenitor zone (*Figure 3E*, blue line). Therefore, unlike the steep gradient in probability of *sygl-1* transcriptional activation, the mean number of transcripts per ATS was not graded. To gauge the full *sygl-1* output per nucleus, we summed ATS signal intensities in each nucleus and plotted those sums as a function of position (*Figure 3F*). The mean of these summed intensities was graded (*Figure 3F*, blue line), consistent with the gradient in the number of *sygl-1* ATS per nucleus (*Figure 3C*). Therefore, number of nascent transcripts made in response to GLP-1/Notch signaling is graded at a cellular level, but that gradient is based on the graded firing probability at individual loci. We conclude that GLP-1/Notch signaling activates transcription in a graded fashion in the distal *C. elegans* gonad, and that this gradient reflects the probability of transcriptional activation, both in single cells (*Figure 3A*) and individual chromosomal loci (*Figure 3B–D*).

To ask if a transcriptional gradient was similarly created when signaling was unregulated, we assayed a *glp-1* gain-of-function (*gf*) mutant harboring a mutation similar to those of cancer-causing human Notch receptors (*Berry et al., 1997*). The *sygl-1* ATS were not graded in *glp-1(gf)* germlines, but instead had an essentially uniform distribution and were generated at a level similar to that of the wild-type far-distal gonad with respect to both ATS probability and intensity (*Figure 3—figure supplement 1A–D*). Therefore, the unregulated GLP-1/Notch receptor does not increase signaling activity *per se* but instead drives signaling ectopically.

## Gradient of *sygl-1* cytoplasmic mRNAs differs from that of *sygl-1* ATS

To learn where GLP-1/Notch transcriptional activation generates mature mRNAs, we analyzed *sygl-1* mRNAs as a function of position along the distal-proximal axis of the distal gonad (*Figure 3G,H*). Our primary question was whether *sygl-1* cytoplasmic mRNAs were distributed in the same pattern as *sygl-1* ATS. To assess the number of *sygl-1* cytoplasmic mRNAs on a per cell basis, we estimated cell boundaries computationally in 3 dimensions (*Figure 3—figure supplement 2A*) and recorded mRNAs within those boundaries. Most *sygl-1* mRNAs were restricted to a region of 1–45 µm (~1–10 gcd) from the distal end (*Figure 3G*, dashed purple line). Moreover, virtually all cells had *sygl-1* mRNA in the distal-most region (1–20 µm or ~1–5 gcd), with the percentage decreasing more proximally (*Figure 3H*). Therefore, the mRNA pattern was distinct from the ATS pattern, even though both were assessed using the same exon probe and same images. We conclude that *sygl-1* nascent transcripts provide a more direct and accurate readout of GLP-1/Notch activated transcription than *sygl-1* mRNA.

Perhaps most striking were germ cells at the distal end that lacked *sygl-1* ATS but contained *sygl-1* mRNA. One straightforward explanation, consistent with transcriptional pulses (e.g. *Larson et al., 2009*; *Raj and van Oudenaarden, 2009*; *Chubb and Liverpool, 2010*), is that dynamic *sygl-1* ATS produce *sygl-1* mRNAs that persist in the cytoplasm. Alternatively, mRNAs might move between

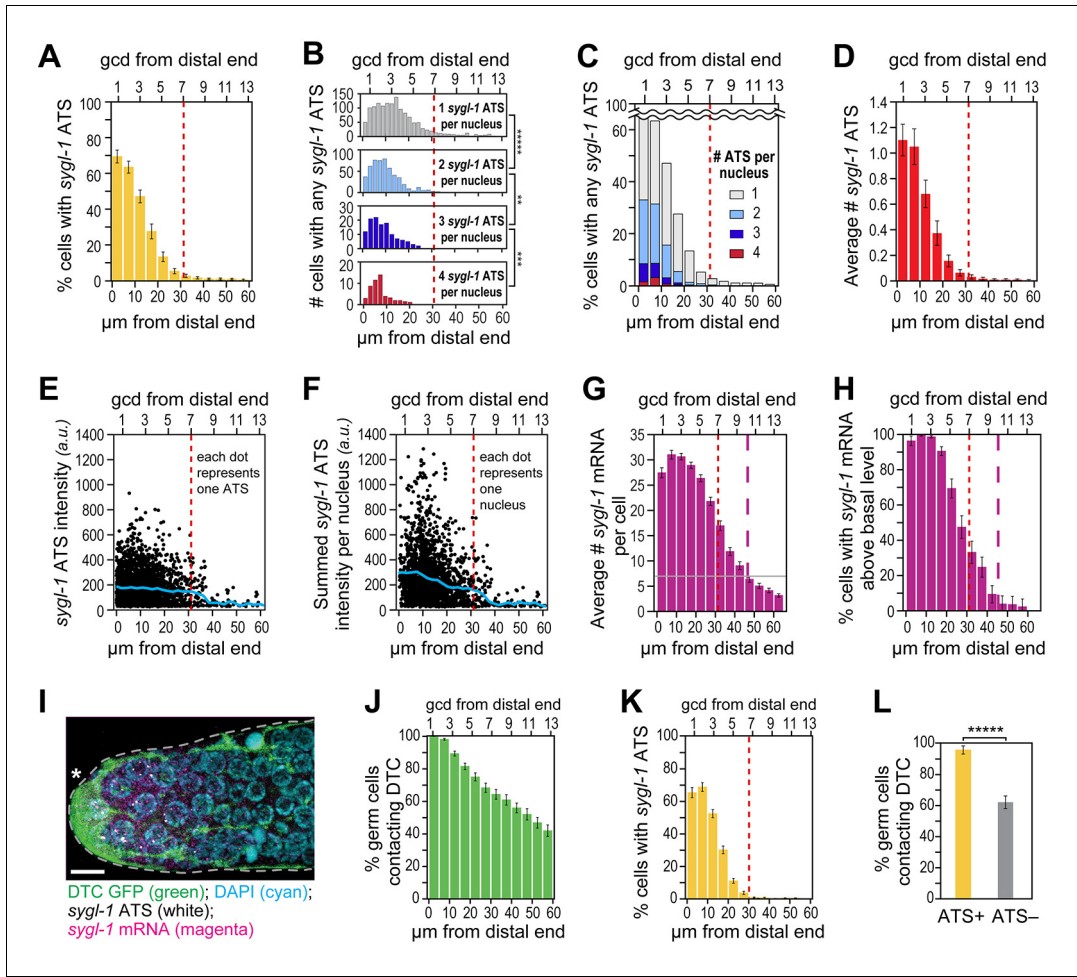

**Figure 3.** The *sygl-1* transcriptional response is spatially graded. (**A–H** and **K,L**) *sygl-1* smFISH signals as a function of position along the distal-proximal axis of adult gonads. Positions were measured at 5 µm (**A,C–K**) or 2 µm (**B**) intervals from the distal end (x-axis below each graph), and translated to the conventional measure of germ cell diameters (gcd) from the distal end (x-axis above each graph). Error bars: standard error of the mean (SEM). n = 78 gonads. Red dashed line marks the site where the mean percentage of ATS-positive germ cells falls lower than 5%; purple dashed line marks the site where the mean percentage of mRNA-positive germ cells falls lower than 5%. (**A**) Gradient in percentage of germ cell nuclei with any number of *sygl-1* ATS as a function of distance from the distal end in the wild-type adult gonad. By contrast, that percentage is essentially uniform in *glp-1(gf)* gonads (**Figure 3—figure supplement 1A**). (**B**) Numbers of cells with one, two, three or four *sygl-1* ATS per nucleus as a function of distance from distal end. Total n = 2058 nuclei. **p<0.01, ***p<0.001, *****p<0.00001 by t-test. (**C**) Percentages of total ATS-positive nuclei that have one, two, three or four *sygl-1* ATS per nucleus as a function of distance from distal end. Total n = 2058 nuclei. (**D**) Average number of *sygl-1* ATS as a function of distance from the distal end. (**E**) Signal intensities of individual *sygl-1* ATS do not change substantially in region of graded ATS (1–7 gcd, border marked with red line). Each dot represents a single *sygl-1* ATS. The blue curve indicates mean ATS intensity. n = 2978 ATS. Signal intensities of individual *sygl-1* ATS are comparable in *glp-1(gf)* gonads (**Figure 3—figure supplement 1B**). (**F**) Summed *sygl-1* ATS intensities per nucleus were used as a measure for total *sygl-1* nascent transcripts per nucleus and then plotted as a function of distance from the distal end, with each dot representing a single nucleus. The blue curve shows the mean for summed ATS intensities at each position relative to the distal end. n = 2058 nuclei. (**G**) The number of *sygl-1* cytoplasmic mRNA per cell as a function of distance from the distal end. Boundaries of cells were estimated using 3-D Voronoi diagram (**Figure 3—figure supplement 2A**; see Materials and methods for details). The grey line marks the average basal *sygl-1* mRNA level, which was calculated by averaging *sygl-1* mRNA density over a region where mRNAs are both least abundant and no longer graded (the 40–60 µm interval in each gonadal image). A cell with abundant *sygl-1* mRNA can reside next to a cell with few mRNA (**Figure 3—figure supplement 2B–E**). (**H**) Percentage of germ cells with *sygl-1* cytoplasmic mRNAs above basal level as a function of distance from the distal end. (**I**) *sygl-1* smFISH in adult distal gonad with DTC and its processes visualized with myristoylated GFP (green) and nuclei seen

*Figure 3 continued*

with DAPI (blue). *sygl-1* exon probes mark cytoplasmic mRNAs (magenta) and intron probes (yellow) highlight ATS. Overlap is in white. Conventions and scale as in **Figure 1E**. Z-projection is shown. (J-L) Quantitative analyses of *sygl-1* response in animals where DTC was marked with myristoylated GFP, n = 60 gonads at 24 hr post mid-L4 stage. Error bar: SEM. (J) Percentage of germ cells in contact with DTC or its processes as a function of distance from the distal end. (K) Percentage of germ cells with one or more *sygl-1* ATS as a function of distance from the distal end. Red dashed line marks region defined as in **Figure 3A**. (L) Percentage of germ cells in contact with DTC or its processes within 30 μm from the distal end. Left bar, ATS-positive cells; right bar, ATS-negative cells or without ATS. *****p<0.00001 by chi-square test for independence.

The following figure supplements are available for figure 3:

**Figure supplement 1.** The *sygl-1* transcriptional response is not spatially graded in *glp-1(oz112gf)* mutant germ cells.

**Figure supplement 2.** Cells with abundant *sygl-1* mRNA can reside next to cells with little mRNA.

---

germ cells given their interconnections via a cytoplasmic core or 'rachis' (**Hirsh et al., 1976**). We were not able to assay the movement of individual mRNAs, because live imaging of RNAs has not yet been achieved in this system. Instead, we asked if neighboring cells had similar mRNA densities and found cells with many *sygl-1* mRNAs residing immediately adjacent to a neighbor with few or no *sygl-1* mRNAs (**Figure 3—figure supplement 2B**). We also assessed mRNA densities in ATS-positive cells, ATS-negative cells and the rachis. The *sygl-1* mRNAs in the rachis are evidence for movement from cells to rachis (**Figure 3—figure supplement 2C**). However, ATS-positive cells had more *sygl-1* mRNAs than did ATS-negative cells, and both had more than the rachis in the distal-most gonad (**Figure 3—figure supplement 2D**). This analysis does not exclude a role for mRNA movement, but suggests that free diffusion of mRNAs between cells through the rachis is unlikely. We favor instead the idea that *sygl-1* transcriptional pulses are of shorter duration than mRNA half-life and that this 'bursty' transcriptional activation generates a nearly uniform field of mRNAs (see Discussion).

## DTC contact correlates with firing of *sygl-1* ATS but is not sufficient

To explore the relationship between the signaling DTC and germ cells activated for *sygl-1* transcription, we analyzed *sygl-1* ATS in gonads with the DTC cellular architecture marked using myristoylated GFP (**Figure 3I**). DTC-germ cell contacts occurred in a gradient that extended far beyond the region with *sygl-1* ATS (**Figure 3J,L**). Virtually all germ cells were in DTC contact at the distal end, ~60% of germ cells remained in contact at 7 gcd from the distal end, and ~40% were still in contact more proximally at 13 gcd from the distal end (**Figure 3**). Yet, as in wild-type animals lacking GFP, *sygl-1* ATS were restricted to a region of 1–30 μm (~1–7 gcd) from the distal end and the percentage of cells with *sygl-1* ATS was graded (**Figure 3K**). Therefore, DTC contacts were much more extensive than the Notch-dependent ATS. Most importantly, the DTC was in contact with nearly all ATS-positive cells as well as with many ATS-negative cells (**Figure 3L**). We conclude that DTC contact is likely necessary for Notch-dependent transcriptional activation, but that it does not ensure Notch-dependent transcriptional activation at any given time.

## *lst-1* smFISH confirms the probabilistic response to Notch signaling

To investigate Notch-regulated transcriptional activation of a second gene, we assayed *lst-1* ATS and mRNAs (**Figure 4A**) and found essentially the same features seen with *sygl-1*: *lst-1* ATS were detected in only some germ cell nuclei at the distal end, whereas *lst-1* mRNAs were in most of them (**Figure 4B**); most *lst-1* ATS-positive nuclei had only one ATS but some had two, three or even four (**Figure 4C**); and pairwise comparisons of ATS intensities within a single nucleus revealed little correlation (**Figure 4D**). Therefore, like *sygl-1,* individual *lst-1* genes are activated independently.

We next used differentially-labeled probes to image *sygl-1* and *lst-1* ATS in the same gonad. Some nuclei had *sygl-1* but not *lst-1* ATS (**Figure 4F**, arrow; **Figure 4G,H**); others had *lst-1* but not *sygl-1* ATS (**Figure 4F**, arrowhead; **Figure 4G,H**); and others had either both or neither. Within a single nucleus, the numbers of *sygl-1* and *lst-1* ATS and their summed ATS intensities were not

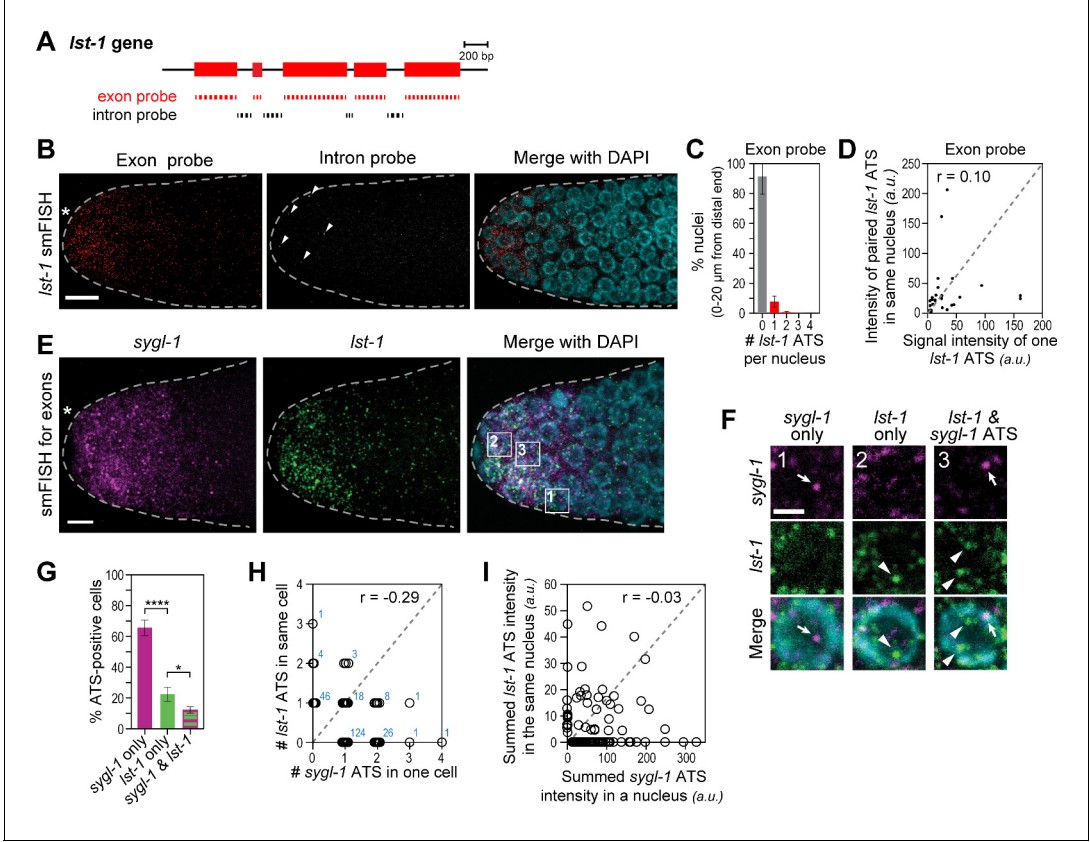

**Figure 4.** The *lst-1* transcriptional response to Notch signaling. (**A**) Schematic of *lst-1* exon/intron structure. Exon-specific (red) and intron-specific (black) smFISH probes were labeled with different fluors (see Materials and methods). (**B**) *lst-1* smFISH in distal gonad. Exon probes (red); intron probes (white). DAPI marks nuclei (blue). Arrowheads indicate ATS. Conventions and scale as in *Figure 1E*. (**C**) Percentage of nuclei with 0–4 *lst-1* ATS per nucleus, calculated from 2107 nuclei in 36 gonads: 162 nuclei had one ATS, 15 had two ATS, 3 had three ATS and 2 had four ATS. Error bars: standard deviation. (**D**) Pair-wise comparisons of *lst-1* ATS intensities within one nucleus (total of 30 ATS from 36 gonads). Each black dot represents one pairing. Grey dashed line indicates a perfect correlation (Pearson's correlation coefficient r = 1); r indicates correlation coefficient from data. (**E**) Double-labeled smFISH against *sygl-1* (magenta) and *lst-1* (green) exons using distinct fluors (see Materials and methods). Conventions and scale as in *Figure 1E*. Full Z-projection is shown. (**F**) 10X magnification of nuclei within boxes in *Figure 4E*, right panel. Each panel shows a restricted Z-projection that only includes the corresponding nucleus. Arrow: *sygl-1* ATS; arrowhead: *lst-1* ATS. Scale bar: 2 μm. (**G**) Percentage of cells with *sygl-1* ATS only, *lst-1* ATS only or both *sygl-1* and *lst-1* ATS, out of all ATS-positive cells identified (n = 233 cells from 15 gonads). ****p-value<0.0001 and *p<0.05 by t-test. (**H**) Plot of nuclei possessing both *sygl-1* and *lst-1* ATS, with each open circle representing one nucleus (n = 233; 15 gonads). Overlapping data points (open circles) are spread using 'jitter' function in MATLAB. Blue numbers show how many nuclei of each type were found. For example, three nuclei had two *lst-1* ATS and one *sygl-1* ATS. Pearson's correlation coefficient (r) is shown on top. Grey dashed line indicates a perfect correlation (r = 1). (**I**) Comparison of summed ATS intensities of *sygl-1* and *lst-1* within the same nucleus (data from 128 nuclei in 9 gonads). Each open circle represents a nucleus. r indicates Pearson's correlation coefficient from data. Grey dashed line indicates a perfect correlation (r = 1).

correlated (*Figure 4H,I*). Therefore, at any given time, GLP-1/Notch can activate transcription of *sygl-1* only, *lst-1* only, both loci or neither. This finding with a second gene strongly supports the conclusion that Notch activates transcription independently at individual chromosomal loci.

## Notch NICD strength determines *sygl-1* transcriptional activity

To gain insight into the mechanism underlying the Notch-regulated ATS gradient, we investigated the effect of Notch NICD strength on ATS generation. The NICD is a key component of the Notch-dependent transcription complex (*Figure 5A*). Here we focus on a temperature-sensitive Notch receptor mutant, *glp-1(q224)*, that behaves like a null at the restrictive temperature (25°C) but is at least partially functional at the permissive temperature (15°C) (*Austin and Kimble, 1987*). Molecularly, GLP-1(q224) harbors a missense mutation that renders its NICD temperature sensitive for assembly into the Notch-dependent transcription complex (*Figure 5A*) (*Kodoyianni et al., 1992*;

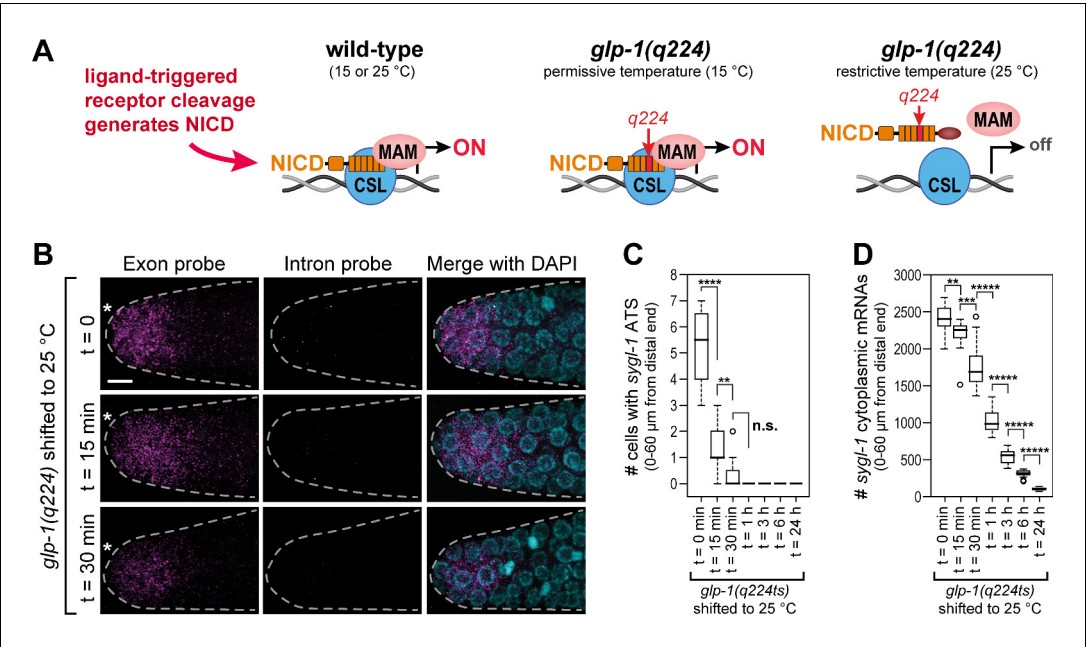

**Figure 5.** The *sygl-1* transcriptional response is abolished in *glp-1(q224)* mutant germ cells at the restrictive temperature. (**A**) Schematics of GLP-1/Notch-dependent nuclear complex in wild-type and *glp-1(q224)* temperature-sensitive mutant. Left, wild-type GLP-1/Notch intracellular domain (NICD) is cleaved from receptor and assembles into a complex in the nucleus to activate transcription; middle, *glp-1(q224)* NICD has a missense mutation in its fourth ankyrin repeat (red) but can assemble into the complex at the permissive temperature; right, *glp-1(q224)* NICD fails to assemble into the complex at the restrictive temperature. Effect of *glp-1(q224)* on complex formation from *Petcherski and Kimble (2000)*. CSL, Notch-specific DNA-binding protein; MAM, mastermind-like transcriptional coactivator. (**B-D**) *glp-1(q224)* homozygous adults were raised at the permissive temperature (15°C) and then shifted to the restrictive temperature (25°C) for defined time intervals. Dissected gonads were probed for *sygl-1* transcripts with smFISH. n ≥ 20 gonads for each time point. The wild type *sygl-1* transcriptional response is essentially the same at 15, 20 and 25°C (*Figure 5—figure supplement 1*). (**B**) Z-projections after shift to the restrictive temperature for 0 min (top), 15 min (middle) and 30 min (bottom). *sygl-1* exon probes (magenta); *sygl-1* intron probes (yellow); and DAPI marks nuclei (blue). Conventions and scale as in *Figure 1E*. (**C,D**) smFISH signals for *sygl-1* ATS (**C**) or mRNAs (**D**) were analyzed in all cells in the region of 0–60 µm (1–13 gcd) from distal end. Asterisks indicate p-value range by t-test. *p<0.05; **p<0.01; ***p<0.001; ****p<0.0001; *****p<0.00001; n.s.: not significant (p>0.05).

The following figure supplement is available for figure 5:

**Figure supplement 1.** Wild-type *sygl-1* transcriptional response is essentially the same at 15, 20 or 25°C.

---

*Petcherski and Kimble, 2000*). The wild-type *sygl-1* transcriptional response was indistinguishable in animals raised at 15, 20 and 25°C (*Figure 5—figure supplement 1*). By contrast, *glp-1(q224)* mutants raised at 15°C and shifted to 25°C as adults lost most *sygl-1* ATS within 15 min after the shift (*Figure 5B,C*); *sygl-1* mRNAs were lost more slowly but most were gone within three hours (*Figure 5B,D*). From this data, we estimate the half-life of GLP-1(q224)-dependent ATS to be <15 min and *sygl-1* mRNAs to be ~1 hr. We conclude that GLP-1(q224) is incapable of activating transcription in nematode germ cells at the restrictive temperature.

Phenotypically, the GLP-1(q224) receptor appears weaker than the wild-type receptor at the permissive temperature: *glp-1(q224)* progenitor zones are smaller than normal at 15°C (*Figure 6A*) (*Cinquin et al., 2010*; *Fox and Schedl, 2015*). To ask if the GLP-1(q224) receptor activates transcription more weakly than wild-type, we compared *sygl-1* ATS in wild type and *glp-1 (q224)* germ cells. The percentage of germ cell nuclei with *sygl-1* ATS was about two-fold lower in *glp-1(q224)* mutants than in wild type (compare *Figures 3A* and *6B*), mean intensity of *sygl-1* ATS

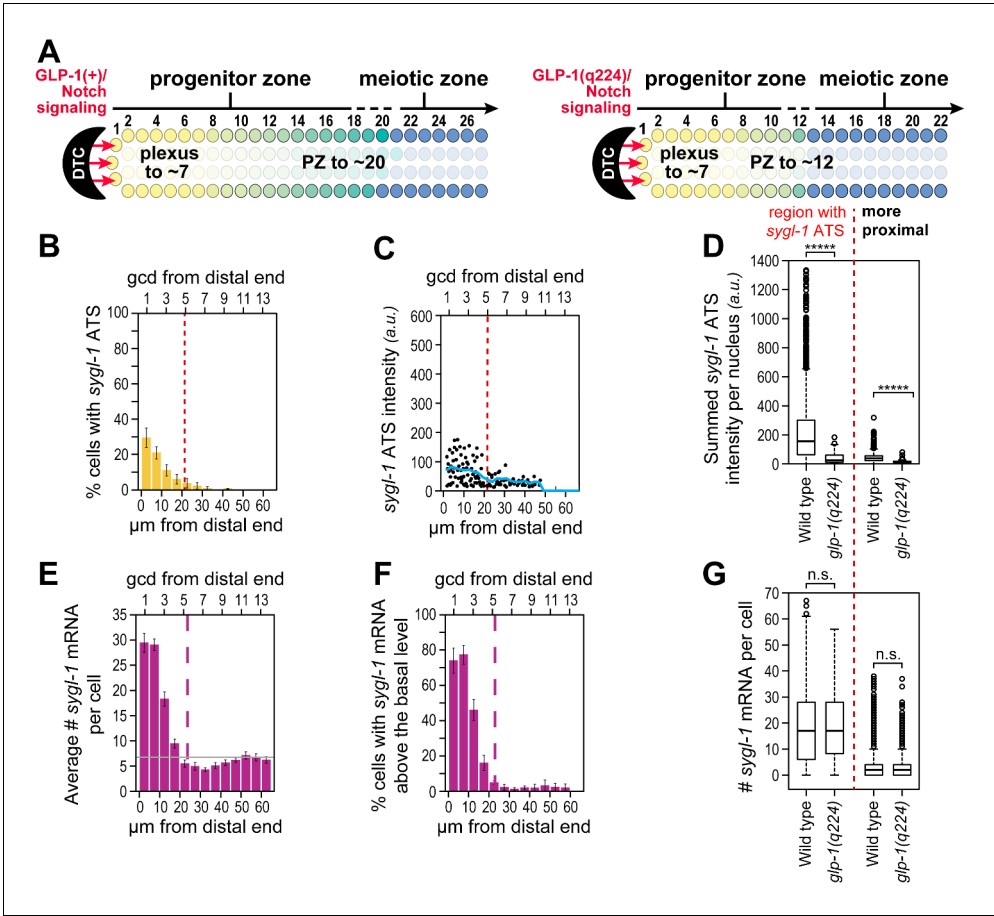

**Figure 6.** Weakened *sygl-1* transcriptional response in *glp-1(q224)* at permissive temperature. (**A**) The *glp-1(q224)* progenitor zone is smaller than wild-type (*Cinquin et al., 2010*; *Fox and Schedl, 2015*) though its plexus of DTC processes is essentially the same (*Byrd et al., 2014*). (**B,C,E,F**) *sygl-1* transcriptional response in *glp-1(q224)* at 15°C (n = 20 gonads). Format, conventions and scales as in *Figure 3A,E,G,H*. (**B**) Percent cells with *sygl-1* ATS as a function of position. (**C**) Individual *sygl-1* ATS intensities as a function of position. (**D**) Summed *sygl-1* ATS intensities per nucleus compared between wild type and *glp-1(q224)*. Left of red line, comparison from region with graded *sygl-1* ATS, which for wild-type was 0–30 μm and for *glp-1(q224)* was 0–20 μm; right of red line, comparison from region proximal to graded *sygl-1* ATS, which for wild-type was 30–60 μm and for *glp-1(q224)* was 20–60 μm. *****p<0.00001 by t-test. (**E**) Average number *sygl-1* mRNAs per cell as a function of position. The grey line marks the average basal *sygl-1* mRNA level (see Materials and methods). (**F**) Percent cells with *sygl-1* mRNAs as a function of position. (**G**) Number of *sygl-1* mRNAs per cell compared between wild type and *glp-1(q224)*. Regions compared were same as described for *Figure 6D*. n.s.: not significant by t-test.

was also roughly two-fold lower (compare *Figures 3E* and *6C*, *blue lines*), and means of the summed intensities of *sygl-1* ATS per nucleus were even more than two-fold lower in the mutant (*Figure 6D*). In addition, the percentage of cells with *sygl-1* mRNA was lower in *glp-1(q224)* mutants than in wild type (compare *Figures 3G* and *6E*). We used this lower percentage in the mutants to show that cells with abundant *sygl-1* mRNA could reside next to neighbors with few or no *sygl-1* mRNAs, even at the distal-most end of the gonad (*Figure 3—figure supplement 2E*). Intriguingly, mRNA number per cell was comparable to wild type in the distal-most cells (*Figure 6F,G*). Because *glp-1(q224)* harbors a mutation in its NICD, we conclude that NICD strength regulates both probability and intensity of transcriptional activation.

## Testing models to explain graded Notch-dependent ATS

Notch-dependent *sygl-1* ATS are graded in both wild type (*Figure 3A*, *Figure 3K*) and *glp-1(q224)* mutants at the permissive temperature (*Figure 6B*). We considered two explanations of that graded pattern. One idea was that Notch activity might be graded (*Figure 7A*, left). For example, the abundance or activity of Notch ligand might be graded with most in the DTC cell body and less in DTC processes, but we emphasize that activity of any pathway component might serve the same end. A second idea is that signaling is localized to the distal-most end but that the stability of the Notch transcription complex decays as germ cells move proximally within the niche (*Figure 7A*, right). One way to distinguish between these two models is to measure abundance, half-life and activity of each pathway component according to position. However, this approach is challenging, in part because many Notch ligands exist in this system and in part because activity assays are not available for most of the components. Instead, we tested predictions of the two models, which are vastly different (*Figure 7B*). The first model predicts establishment of the graded pattern at the rate of ATS generation, and the second predicts its establishment at the rate of germ cell movement within the niche. What are those rates? Neither is known with any precision, but rough estimates can be made. In diverse eukaryotic cells, transcriptional elongation occurs at a rate of roughly 1–5 kb/min (e.g. *Ben-Ari et al., 2010*; *Danko et al., 2013*). If the *sygl-1* elongation rate falls in a similar range, *sygl-1* ATS should be generated relatively quickly, certainly within an hour. By contrast, germ cell movement along the distal-proximal axis is relatively slow, on the order of one cell diameter per hour

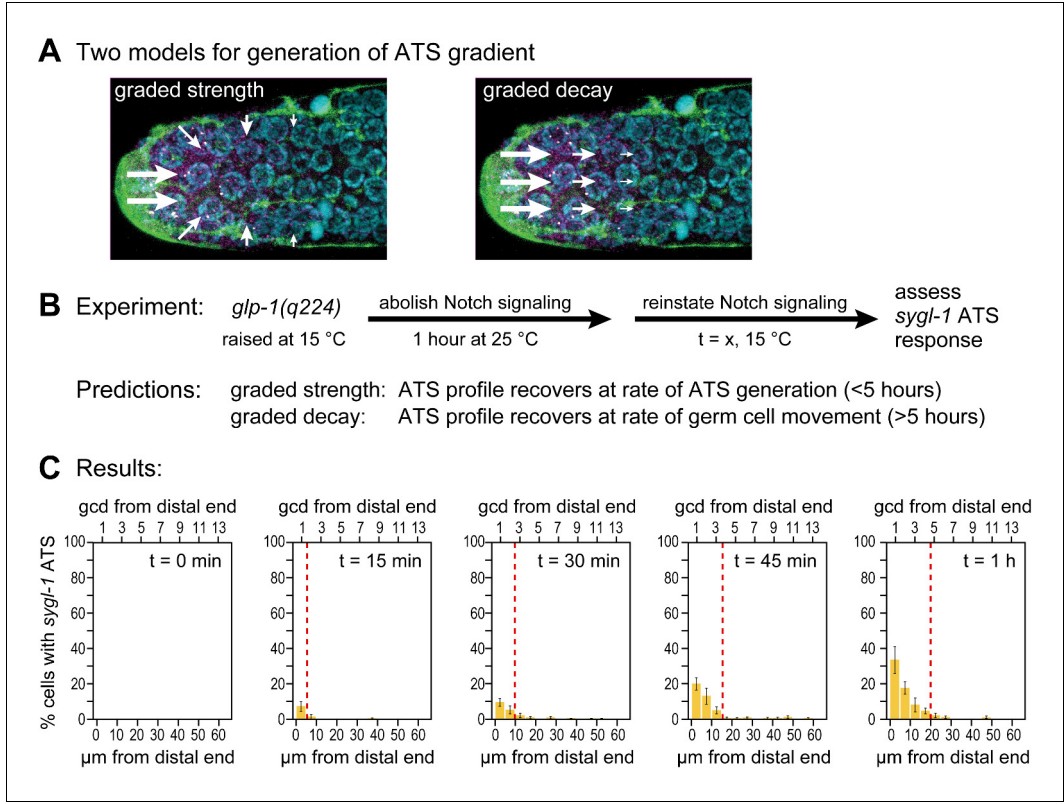

**Figure 7.** Graded *sygl-1* transcriptional response likely reflects graded signaling strength. (**A**) Two models to generate graded *sygl-1* transcriptional response. Left, DTC signaling is graded within the niche; right, signaling is primarily at distal end but decays as cells move proximally within the niche. (**B**) Above, experiment to monitor rate of ATS pattern establishment after being abolished. Below, prediction made by each model. (**C**) Graded response is re-established within 1 hr. Percentage germ cells with *sygl-1* ATS as a function of distance at 15 min intervals during reformation. Red line marks the region with *sygl-1* ATS, as in *Figure 3A*. Conventions as in *Figure 3A*; n = 21 gonads for each time point.

(*Crittenden et al., 2006*; *Cinquin et al., 2015*) or several hours to traverse the region with graded *sygl-1* ATS.

To test these predictions, we measured the rate at which the graded pattern of *sygl-1* ATS was re-established after being fully abolished. Specifically, we shifted *glp-1(q224)* mutants to the restrictive temperature for one hour, well beyond the time when ATS are no longer detected (*Figure 5C*), then shifted them back to the permissive temperature to reinstate signaling (*Figure 7B*) and finally assayed *sygl-1* ATS at timed intervals (*Figure 7C*). The answer was clear: the *sygl-1* ATS pattern was fully re-formed within an hour. Both the percentage of nuclei with *sygl-1* ATS and their spatial extent were the same as without any shift (*Figure 7C*, compare to *Figure 6B*). Therefore, the mechanism underlying the spatially graded pattern of Notch-dependent ATS is not decay of the Notch response in germ cells as they move, but instead relies on the spatially graded Notch strength (see Discussion). We conclude that germline stem cells within their niche are subject to spatially modulated Notch signaling.

## Discussion

This work reports a high resolution and quantitative view of Notch-regulated transcriptional activation at endogenous genes in their natural context, a first for any canonical signaling pathway to our knowledge. We find that Notch does not orchestrate a synchronous nuclear response but instead tunes the probability of transcriptional firing and number of nascent transcripts produced. Consistent with that conclusion, Notch activates transcription independently at individual chromosomal loci. We also overturn expectations that the pattern of nuclear active transcription sites (ATS) is essentially the same as the pattern of cytoplasmic mRNAs. ATS are generated in a steep probability gradient across the stem cell pool, but mRNAs are more uniform in the same region and extend beyond it. We also find that NICD strength determines both probability of transcriptional firing and number of nascent transcripts per ATS, providing a new metric for Notch strength. Below we place these insights in context and discuss their broader implications.

### Notch regulates the probability of transcriptional firing

We find that GLP-1/Notch regulates the probability of transcriptional firing at two target genes, *sygl-1* and *lst-1,* and does so independently at individual chromosomal loci. For both genes, Notch activates transcription most commonly at only one chromosomal locus, though in some nuclei, both loci are activated and in others, three or even all four loci can fire in this proliferating population. When both *sygl-1* and *lst-1* are assayed in the same cells, these genes too are activated independently: some nuclei activate only *sygl-1*, others activate only *lst-1*, and some activate both or neither. The possession of any one Notch-activated ATS marks that cell as having received the Notch signal, while absence of other Notch-activated ATS in the same cell confirms the probabilistic nature of firing. Importantly, the existence of ATS-negative nuclei is unlikely a problem with detection, because cytoplasmic mRNAs, which are significantly dimmer than ATS, are seen in the same cells.

Transcriptional bursting has emerged as a general phenomenon in both prokaryotic and eukaryotic cells (e.g. *Elowitz et al., 2002*; *Chubb et al., 2006*; *Raj et al., 2006*; *Boettiger and Levine, 2009*), and a few studies revealed transcriptional bursting in response to signaling. Thus, cAMP modulates transcriptional burst frequency in *Dictyostelium* (*Corrigan and Chubb, 2014*), and MAPK modulates transcriptional burst frequency in tissue culture cells (*Senecal et al., 2014*). We suggest that the sporadic appearance of Notch-responsive ATS is best explained by transcriptional 'bursts' or 'pulses' separated by periods of quiescence (*Figure 8A*). We do not yet know if Notch regulates frequency or duration of these pulses, or if timing of Notch-dependent transcriptional activation follows a set rhythm once initiated. Answers to these questions must await live imaging of nascent RNAs. Regardless, our results show that Notch regulates the probability of transcription firing at endogenous genes in a wild-type intact tissue, which provides a critical foundation for future investigations with live imaging, which is expected to have lower resolution and typically relies on manipulated reporters.

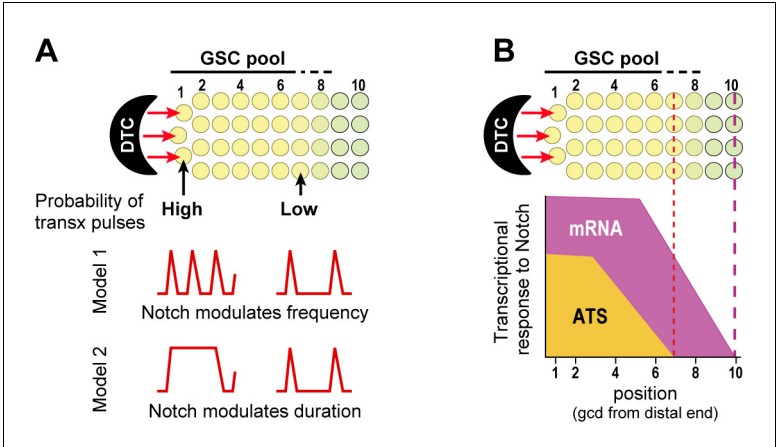

**Figure 8.** Gradient of Notch-dependent active transcription sites: models for underlying mechanism (**A**) and relation to mRNA distribution (**B**). (**A,B**) Top, Notch signaling from the DTC niche maintains a pool of germline stem cells (GSCs) (***Cinquin et al., 2010***). (**A**) Notch signaling generates a gradient in the probability of transcriptional activation across the stem cell pool. ATS probability differences may result from modulation of ATS frequency (Model 1) or duration (Model 2). Red lines illustrate theoretical effects on transcriptional bursting at high (left) or low (right) probability. (**B**) Notch signaling from niche generates differently-shaped gradients of ATS and mRNA: mRNA abundance is essentially ungraded in the region where ATS probability is graded; the mRNA gradient extends further proximally than the ATS gradient; and more cells have mRNAs than ATS at any given position.

## NICD strength modulates both probability and intensity of transcription

We compared nuclear responses to wild-type and mutant GLP-1/Notch receptors and found clear differences in their ability to activate transcription. The receptors tested differ at a single amino acid in the GLP-1/Notch intracellular domain (NICD), which is integral to the Notch transcription complex (***Kovall and Hendrickson, 2004***). In yeast and in vitro, the mutant NICD is incapable of assembly into the Notch transcription complex at a higher temperature and less capable of assembly than the wild-type NICD at a lower temperature (***Petcherski and Kimble, 2000***). Here we show in nematode germ cells that this same mutant NICD is incapable of Notch-activated transcription at the restrictive temperature and weaker than the wild-type NICD at the permissive temperature. The wild-type and mutant NICDs each had a characteristic probability of transcriptional firing and produced a characteristic number of nascent transcripts, with the mutant at the permissive temperature being about two-fold less effective than wild-type. The simplest interpretation is that the mutant NICD is less capable than wild-type of forming or maintaining the Notch transcription complex in germ cells. The molecular consequence of the mutant's two-fold lower transcriptional response is the generation of a shorter than normal ATS gradient (this work), and its biological consequence is a reduced progenitor zone (***Cinquin et al., 2010***; ***Fox and Schedl, 2015***). The discovery that NICD strength modulates both probability and intensity of the Notch transcriptional response underscores the importance of the NICD for fine-tuning the Notch response, with implications for pattern formation and disease. This finding also highlights the power of a high-resolution ATS assay for assessing molecular consequences of any condition (e.g. defects in any component in the pathway as well as physiological or environmental factors) that affect Notch transcriptional activation.

## A GLP-1/Notch-regulated gradient in ATS probability across the stem cell pool

GLP-1/Notch signaling from the DTC niche not only maintains a pool of germline stem cells but also establishes polarity and patterning in the germline tissue (***Kimble and White 1981***; ***Austin and Kimble, 1987***; ***Lander et al., 2012***). Yet the spatial extent of GLP-1/Notch influence was poorly understood prior to this work. We now learn that GLP-1/Notch activates transcription over a defined

region (7 germ cell diameters from the distal end) and that its nuclear response is steeply graded over that region. Importantly, GLP-1/Notch forms a gradient of ATS probability rather than ATS size, and to our knowledge, no gradient of this type has been seen before. We imagine that Notch generates this ATS probability gradient by tuning either the frequency or duration of transcriptional pulses (*Figure 8A*), possibilities that can be distinguished once live imaging of nascent transcripts has been established.

One mechanism for generating the ATS probability gradient might have been that Notch signaling was restricted to the distal end and that the gradient formed by gradual decay of the Notch transcription complex as germ cells moved proximally within the niche. However, after abolishing the Notch nuclear response, recovery of the gradient was too fast to support such a mechanism (*Figure 7*). We favor instead a mechanism that establishes the graded activity of the Notch transcription complex. Possibilities include graded changes in ligand abundance or activity, graded changes in NICD abundance or activity or graded changes in chromatin modifications, any of which could play a role in forming the ATS probability gradient. Regardless, we suggest that graded loss of active transcription sites at loci encoding key stem cell regulators is likely the molecular basis of the long sought 'initial trigger' for stem cells to begin their transition towards a differentiated state and, as such, this gradient is likely to be critical for establishing size of the stem cell pool.

## ATS provide an accurate assay of productive notch activity

Cytoplasmic mRNAs provide a commonly used readout of the Notch transcriptional response and clearly their distribution is crucial biologically. In the *C. elegans* distal gonad for example, Notch-dependent mRNAs generate two key stem cell regulators, SYGL-1 and LST-1, that maintain a pool of stem cells with apparently equivalent developmental potential (*Kershner et al., 2014*). Yet the distributions of Notch-dependent ATS and mRNAs are not the same. At any given position, more germ cells had *sygl-1* and *lst-1* mRNAs than had *sygl-1* and *lst-1* ATS, and mRNAs extended more proximally than ATS (*Figure 8B*). This difference is unlikely to be a detection problem, because the smFISH signal for mRNA was less intense than that for ATS while mRNAs were more broadly distributed than ATS.

One simple explanation for the ATS *vs.* mRNA difference is that Notch-dependent ATS may be less stable than Notch-dependent mRNAs. Consistent with that idea, we estimate half-life of *sygl-1* nascent transcripts (detected at nuclear ATS) to be <15 min and half-life of *sygl-1* mRNAs to be ~1 hr (both estimates based on rate of loss after Notch removal, *Figure 5C,D*). Therefore, Notch-dependent ATS are a more accurate assay of Notch activity. We do not know yet whether Notch-dependent ATS are more accurate than presence of nuclear NICD, another marker of active Notch signaling (*Del Monte et al., 2007*). We note however that Notch-dependent ATS reveal transcriptional activation, the ultimate test of productive Notch signaling, and that ATS also provide a quantitative readout of signaling strength.

## Summary and future directions

Our findings offer a new window into the Notch transcriptional response and demonstrate the importance of assaying nascent transcripts at active transcription sites as the most accurate readout for canonical signaling. The *C. elegans* gonad was uniquely suited for this analysis because of its defined signaling cell and continuous activity across a field of many receiving cells, which offered statistical power for quantitating the probabilistic nuclear response. We suspect that all canonical signaling pathways will follow guidelines similar to those found here for Notch. Key questions for the future include how individual components of canonical signaling pathways affect the probability and intensity of active transcription sites, and ultimately how environment, pathogenesis and aging impact their control. Precise molecular answers to these fundamental questions are now accessible.

## Materials and methods

### Nematode culture

All strains were maintained at 20°C as described (*Brenner, 1974*), except *glp-1(q224)* (*Austin and Kimble, 1987*) which was maintained at 15°C. The wild type was N2 Bristol. Mutations were as follows: *LG I: gld-1(q485)* (*Francis et al., 1995*); *gld-2(q497)* (*Kadyk and Kimble, 1998*); *rrf-1(pk1417)*

(*Sijen et al., 2001*); LG III: *glp-1(q46)* (*Austin and Kimble, 1987*); *glp-1(q224)* (*Austin and Kimble, 1987*); and *glp-1(oz112 gf)* (*Berry et al., 1997*). The balancer used for *LG I* was *hT2[qIs48]* (*Siegfried and Kimble, 2002*). Transgene was as follows: *LG III: qIs153[P_{lag-2}::MYR::GFP; P_{ttx-3}:: DsRED]* (*Byrd et al., 2014*). For smFISH and antibody staining of the wild-type strain, animals were grown at 20°C until 24 hr post-mid-L4 stage (adult), 15°C until 36 hr post-mid-L4 stage (adult), or 25°C until 12 hr post-mid-L4 stage (adult). *glp-1(q224)* animals were grown at 15°C to 36 hr post mid-L4 stage (adult) and then shifted to 25°C (restrictive condition) to block GLP-1/Notch activity. L4 worms were processed for smFISH and/or DAPI staining for the following animals: *gld-2(q497) gld-1 (q485)* double mutants, *gld-2(q497) gld-1(q485); glp-1(q46)* triple mutants and *glp-1(oz112 gf)* single mutants.

## Nematode strains and reagents used in this study

N2: wild type
BS860: *unc-32(e189) glp-1(oz112 gf)/ dpy-19(e1259) glp-1(q172) III*
JK1107: *glp-1(q224ts) III*
JK3545: *gld-2(q497) gld-1(q485) I/ hT2[qIs48](I;III); unc-32(e189) glp-1(q46) III/ hT2[qIs48](I;III)*
JK4533: *qSi153[P_{lag-2}::myr-GFP; Pttx-3::DsRED] III*
JK4862: *glp-1(q46) III/ hT2[qIs48](I;III)*
JK4873: *gld-2(q497) gld-1(q485) I/ hT2[qIs48](I;III); unc-32(e189) III/ hT2[qIs48](I;III)*
NL2098: *rrf-1(pk1417) I*

## Single-molecule RNA fluorescence *in situ* hybridization (smFISH)

Details of our smFISH method can be found at Bio-protocol (*Lee et al., 2017*). Custom Stellaris FISH probes (Biosearch Technologies, Inc., Petaluma, CA) were designed against the exons and introns of *sygl-1* or *lst-1* by utilizing the Stellaris FISH Probe Designer available online at www.biosearchtech. com/stellarisdesigner. The *sygl-1* exon-specific probe set includes 31 unique oligonucleotides labeled with CAL Fluor Red 610. The *sygl-1* intron-specific probe set includes 48 unique oligonucleotides tagged with Quasar 570. The *lst-1* exon-specific probe set includes 47 unique oligonucleotides labeled with CAL Fluor Red 610. The *lst-1* intron-specific probe set includes 26 unique oligonucleotides tagged with Quasar 670. The *let-858* intron-specific probe set includes 48 unique oligonucleotides tagged with CAL Fluor Red 610. Extruded gonads were hybridized with FISH probes. Briefly, probes were dissolved in RNase-free TE buffer (10 mM Tris-HCl, 1 mM EDTA, pH 8.0) to create a 250 µM probe stock. Synchronized L1 larvae (*Kershner et al., 2014*) were grown on OP50 until 24 hr post mid-L4 stage, unless otherwise noted. Animals were washed off plates with M9 buffer (3 g KH$_2$PO$_4$, 6 g Na$_2$HPO$_4$, 5 g NaCl, 1 ml 1 M MgSO$_4$, 0.1% Tween-20 in 1 L H$_2$O) and dissected in M9/0.25 mM levamisole to extrude gonads. Samples were fixed with 3.7% formaldehyde in 1X PBS with 0.1% Tween-20 at room temperature (RT) for 15–45 min, with gentle agitation. Samples were pelleted at 2000 RPM for 30 s unless noted otherwise. After fixation, samples were washed with PBSTw (1X PBS with 0.1% Tween-20). Samples were permeabilized with PBSTw containing 0.1% Triton X-100 for 10 min at RT, washed twice with PBSTw, resuspended in 70% ethanol, and stored overnight at 4°C. Ethanol was removed and samples incubated in wash buffer (2X SSC, 10% deionized formamide in nuclease-free water) for 5 min at RT. Gonads were hybridized with 0.25 µM *sygl-1* probes or 0.1–0.5 µM *lst-1* probes in hybridization solution (228 mM Dextran sulfate, 2X SSC, 10% deionized formamide in nuclease-free water) overnight at 37°C with rotation. After probe addition, samples were kept in the dark for all incubations and washes. Samples were rinsed once with wash buffer at RT, then incubated in wash buffer for 30 min at 37°C with gentle agitation, followed by two short washes at RT. DNA was labeled by incubation in wash buffer containing 1 µg/mL diamidinophenylindole (DAPI) for 30 min at 37°C. This was followed by two washes at RT. Antibody incubation, if necessary, was performed at this step (see: *LMN-1 visualization with smFISH*). Finally, samples were resuspended in 10 µL Antifade Prolong Gold mounting medium (Life Technologies Corporation, Carlsbad, CA) and mounted on glass slides.

## RNA interference (RNAi) and α-amanitin treatment for smFISH

RNAi feeding experiments were performed in *rrf-1(pk1417)* worms following established protocols (*Ahringer, 2006*). HT115 bacteria containing *sygl-1* or empty (pL4440) RNAi vectors were grown

separately in overnight cultures and then seeded to RNAi plates in equal volume (60 µL). Three L4 larvae (P$_0$) were plated onto RNAi plates and grown until F$_1$ larvae reached 24 hr post mid-L4 stage. For α-amanitin treatment, synchronized L1 larvae were grown to 24 hr post mid-L4 stage. The worms were washed off the plate with M9 and incubated in 500 µL M9/100 µg/mL α-amanitin for 2 hr on a spinning rotor. Following treatment, worms were washed three times with M9. RNAi or α-amanitin treated worms were dissected and prepared for smFISH as described.

## Confocal microscopy setup and image acquisition

Gonads were imaged using a Leica TCS SP8 (confocal laser scanner) equipped with a Leica HC PL APO CS2 63x/1.40 NA oil immersion objective, sensitive hybrid detectors (HyDs) and standard Photomultipliers (PMTs). LAS AF 3.3.1 acquisition software (Leica Microsystems Inc., Buffalo Grove, IL) was used to acquire all images. All gonads were imaged completely (depth >15 µm) with a Z-step size of 0.3 µm. All imaging was done with HyDs except DAPI and DIC, which were imaged on PMTs. Channels were imaged sequentially to avoid bleed-through. For generating figures, all images were processed with linear contrast enhancement in ImageJ 1.49 p using a minimum contrast of 1.05X mean background signal intensity (>300,000 random sampling inside gonad at a random Z-position) and a maximum contrast of 1.25X maximum signal intensity.

For smFISH, the laser scan was bidirectional at 400 Hz and 300% zoom. The *sygl-1* exon probe (CAL Fluor Red 610) was excited at 594 nm (1.2%, HeNe) and signal was acquired at 598–699 nm (gain 40) with a pinhole at 95.5 µm. The *sygl-1* intron probe (Quasar 570) was excited at 561 nm (1.2%, DPSS) and signal was acquired at 565–620 nm (gain 40) with a pinhole at 124.2 µm. The *lst-1* exon probe (CAL Fluor Red 610) was excited at 594 nm (2.75%, HeNe) and signal was acquired at 598–633 nm (gain 40) with a pinhole at 95.5 µm. The *lst-1* intron probe (Quasar 670) was excited at 633 nm (2.75%, HeNe) and signal was acquired at 650–700 nm (gain 40) with a pinhole at 105.1 µm. GFP was excited at 488 nm (1%, Argon) and signal was acquired at 492–555 nm (gain 50). DAPI was excited at 405 nm (0.1–1%, UV) and signal was acquired at 412–508 nm (gain 600–700). Both pinholes were 143.4 µm. A line average of 6–8 with 1–2 frame accumulation was used for all channels except for DAPI, which was with 3–4 line average.

## Detection of nuclei/RNA coordinate reconstruction

All processes were implemented and automated using custom MATLAB codes (*Source code 1*). Matlab 2015a with the 'image processing' toolbox was used for all image processing and analyses. The images acquired in this study were packaged in LIF format and were read in MATLAB with their metadata (*e.g.*pixel size, z-step size and total number of z slices). The X, Y and Z coordinates and signal intensity of gonadal boundaries, nuclei, mRNAs, and ATS (nuclear spots) were recorded as shown in *Figure 2A*. Whole gonadal images (~0–60 µm from the distal end) were quantitated except in *Figure 2E,G,H* (0–30 µm). The gonad boundary was detected using DIC from the central 8 Z-planes. The MATLAB code reoriented the gonads with DTC to the left for further analyses. The gonads were reconstructed in MATLAB as rounded and often not straight cylinders with radii changing along the distal-proximal axis. The gonadal boundaries were used for removing non-specific fluorescent signal located outside the gonad. Next, the nuclei were detected using DAPI signal. In each Z-plane, the function 'imfindcircles' was used to draw concentric nuclear circles to define the nuclear boundary and the threshold for DAPI signal (background level) was determined by 150% of output values of the function 'greythresh' in MATLAB. DAPI signal in each Z-plane was normalized to the mean background intensity within that Z-plane, in a location without nuclei. To generate a 3-D nucleus, a best-fit sphere was drawn to fit concentric circles from different Z-planes. A nucleus was scored as present if (1) concentric circles were identified in ≥4 consecutive Z-planes and (2) the variation of centers within the concentric circles was <0.5 µm in the X,Y direction (Euclidean distance). To estimate nuclear size, the radius of the 3-D spherical nucleus was recorded. To estimate DAPI intensity, the DAPI signal within the 3-D spherical nucleus was summed.

For active transcription site (ATS) detection, image segmentation tools (*e.g.* 'bwconncomp' function) and a Gaussian local peak detection method ('FastPeakFind' function) were used separately to detect ATS-like spots from intron-labeled samples (≥1.5 X brighter than the mean background signal intensity within the working Z-plane). Only ATS spots identified with both detection methods were considered further. Irregularly shaped objects (circularity <0.7) were discarded (*Lee et al. 2015*). The

criteria used to identify ATS candidates were validated by manually comparing detection results to micrographs (training set of 15 gonads). The 'supervised machine learning' was used to determine parameters for ATS determination (e.g. ATS background level, threshold for signal/background ratio). The criteria were that each ATS within a Z-plane must have: (1) a >1.27 (for sygl-1) or >0.8 (for lst-1) ratio of signal to mean overall background in the gonad; (2) a >1.9 (for sygl-1) or >0.9 (for lst-1) ratio of signal to local background (the area located within 3X the distance [~10–17 pixels] from the ATS center); (3) a $<9*10^{-10}$ (for sygl-1) or $<1*10-1$ (for lst-1) p-value in t-tests comparing all pixel values in candidate ATS to its local background; and (4) a size of >3 × 3 pixels (average detected mRNA size is ~4 × 4 pixels) for both sygl-1 and lst-1. After thus scoring ATS candidates, the gonad was 3-D reconstructed and only ATS with an intensity >10 X over mean background were scored as potential ATS. Next, the potential ATS were overlaid with 3-D nuclei in MATLAB. A potential ATS was determined to be a true ATS based on three criteria: (1) RNA spots visualized by sygl-1, lst-1 or let-858 intron-specific smFISH probes were detected inside nuclei; (2) the exon-specific probe detected a nuclear spot, which co-localized with the intron probe (sygl-1 or lst-1); and (3) the signal intensities from both exon and intron probes were at least as bright as a single mRNA (see below).

mRNA spots were detected in a similar way as ATS spots but relied on sygl-1 and lst-1 exon-specific probes. First, nuclear spots, identified with exon-specific probes, were removed. mRNA candidates were identified in each Z-plane using image segmentation tools (as above). Size, volume, shape and fluorescence signal intensity were used to identify and calculate number/intensity of mRNA spots. We calculated mean mRNA intensity and mean mRNA size in each Z-plane. After thus scoring mRNA candidates, the gonad was 3-D reconstructed and any spots <20% of mean mRNA intensity and <20% of mean mRNA size were removed. Using this calculated mean intensity and size, we were able to determine if a single cytoplasmic spot contained one or multiple mRNAs ($\geq 2$ X brighter and larger). The presence of multiple mRNAs in one spot was rare (<5%). The basal mRNA level in each gonad (Figure 3D,E, Figure 6E,F) was calculated by averaging rare dim signals in germ cells located 40–60 μm (~9–13 gcd) from the distal end (in 5 μm sections). mRNA density was calculated as the total number of mRNAs divided by the volume of the region counted (5 μm sections). The gcd (germ cell diameter) was estimated as 4.55 μm from the mean distance of two neighboring nuclei (from 78 wild-type gonads). We confirmed gcd by manually counting the germ cells in maximum Z-projected distal gonads with DAPI-staining. sygl-1 mRNA density in Figure 3—figure supplement 2D was calculated from total number of mRNAs divided by total cytoplasmic volume of ATS-positive or ATS-negative cells, whose boundaries were defined using a Voronoi diagram in 3-D or in the rachis. Cytoplasmic volume excludes nuclear volume. Non-cell core region of gonad at 0–20 μm (4–5 gcd) was used to calculate the volume of the rachis.

Signal intensities for each gonad were determined above. We then set out to normalize exon probes (both ATS and mRNAs) across gonads and strains for direct comparison. Moving forward, we used only the ATS intensities measured from the exon probes for direct comparisons of values between ATS and mRNAs. We set the mean of cytoplasmic mRNA intensity for each 3-D gonad to 10 a.u. (Figure 2C), which enables direct comparison between gonads. To count the number of mRNAs per germ cell, we first determined the boundary of each germ cell using a 3-D Voronoi diagram (Ledoux, 2007; Yan et al., 2010), which draws germ cell boundaries midway between two neighboring nuclei (Figure 3—figure supplement 2A). Germ cell size was restricted to 3 μm from the nucleus center (mean distance from nucleus center to the most distant cell boundary is 3 μm).

To count sygl-1 and/or lst-1 ATS-positive nuclei (Figure 3E–H), smFISH was conducted using both sygl-1 exon probes (31 probes with Quasar 570) and lst-1 exon probes (47 probes with CAL Fluor Red 610). The number of sygl-1 or lst-1 ATS was then manually counted using the 'Cell Counter' plug-in in ImageJ2. The ATS were recorded only when Gaussian-distributed clear nuclear RNA spots were seen on at least three consecutive Z-planes. The results of manual counting were then confirmed by two people looking through the smFISH images on a Z-plane by Z-plane basis. Once counting was done, MATLAB was used to conduct statistical tests and visualize the data.

Statistical analyses of smFISH data included equal variance (Levene's test) and normality tests (using Anderson-Darling Normality test) to determine if assumptions of the ANOVA and t-tests were met. If at least one test did not satisfy requirements (p<0.05), the Kruskal-Wallis test (nonparametric version of ANOVA) or the Kolmogorov-Smirnov (KS) test (nonparametric version of the t-test) was used to compare data.

### Detection of DTC membranes and determination of germ cell boundary

To detect the distal tip cell (DTC) we used JK4533, a transgenic strain expressing myristoylated GFP (marks DTC membranes). A binary image of the GFP signal (DTC mask) on each Z-plane was created. For each Z-plane, staining was considered significant if the GFP signal was $\geq$3 X brighter than the mean background intensity level inside the gonad, similar to ATS detection. The DTC masks were reconstructed in 3-D using MATLAB codes in *Source code 1*. Using this procedure, the DTC body, processes and plexus were reconstructed in detail. Morphology and extent of detected DTC were comparable to a previous study (*Byrd et al., 2014*). *sygl-1* ATS and mRNA were also scored in the same gonad as described above. To estimate the fraction of cells that directly contact the DTC, we overlaid DTC masks with our 3-D Voronoi cells. Cells were considered in contact with the DTC if they had more than 10 pixels (background level) adjacent to a DTC mask.

### LMN-1 visualization with smFISH

Gonads were dissected and smFISH conducted as described above. All reagents were RNase-free. Immunostaining was carried out as described (*Lee et al., 2006*). Briefly, samples were blocked in PBSTw containing 0.5% BSA (PBSB) for 30 min at RT. Fixed gonads were incubated with 1:10 guinea pig LMN-1 (provided by Jun Kelly Liu) in PBSB overnight at 4°C. Samples were washed 3X for 15 min with PBSB and incubated with secondary antibody at 1:500 (Goat anti-guinea-pig AlexaFluor 488, Life Technologies) for 1.5 hr at RT. Samples were washed three times with PBSB and mounted in Antifade Prolong Gold mounting medium.

## Acknowledgements

Many thanks to Sarah Crittenden, Hannah Seidel, Marvin Wickens and Sarah Robinson for critical comments on the manuscript. We also thank Anne Helsley-Marchbanks for help preparing the manuscript and Laura Vanderploeg for help with the figures. Some strains were provided by the CGC, which is funded by NIH Office of Research Infrastructure Programs (P40 OD010440). This work was supported in part by the American Cancer Society - George F Hamel Jr. Fellowship (PF-14–147-01-DDC to EBS). JK is an Investigator of the Howard Hughes Medical Institute.

## Additional information

### Funding

| Funder | Grant reference number | Author |
| --- | --- | --- |
| American Cancer Society | PF-14-147-01-DDC | Erika B Sorensen |
| Howard Hughes Medical Institute | | Judith Kimble |

The funders had no role in study design, data collection and interpretation, or the decision to submit the work for publication.

### Author contributions

CHL, EBS, Conception and design, Acquisition of data, Analysis and interpretation of data, Drafting or revising the article; TRL, Acquisition of data, Analysis and interpretation of data; JK, Conception and design, Analysis and interpretation of data, Drafting or revising the article

### Author ORCIDs

Judith Kimble, http://orcid.org/0000-0001-5622-2073

## Additional files

### Supplementary files

- Source code 1. MATLAB codes for smFISH analysis.

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
