## [Decision Letter]

[Editors’ note: this article was originally rejected after discussions between the reviewers, but the authors were invited to resubmit after an appeal against the decision.]

Thank you for submitting your work entitled "*C. elegans* GLP-1/Notch modulates transcriptional stochasticity in a gradient to maintain a pool of germline stem cells" for consideration by *eLife*. Your article has been reviewed by two peer reviewers, including Eric Haag (Reviewer #2), and the evaluation has been overseen by Janet Rossant as the Senior Editor and Reviewing Editor.

Our decision has been reached after consultation between the reviewers. Based on these discussions and the individual reviews below, we regret to inform you that your work will not be considered further for publication in *eLife*.

The reviewers and the editor all agree that the data presented are of very high quality and add to our understanding of Notch signaling in maintaining germ line stem cells in *C. elegans*. However, the claim of a novel linkage between cell signaling and stochastic transcription was felt not to be fully justified. The crux of your argument was interpreted as: 1) Not all cells are in the same state with respect to transcription, and therefore this means there is a stochastic process at work 2) Because there is nevertheless a clear impact of Notch-GLP-1 signaling on the distribution of transcription, "modulated stochasticity" is a key mechanism of signal transduction. However, mere non-homogeneity of transcription cannot on its own prove that stochastic forces are at work. When nascent transcripts are short-lived, and produced sparsely and asynchronously, many cells will lack them, yet the underlying process could actually be highly regular. In addition, there is now considerable existing evidence for the generality of stochasticity so that it is not clear that your findings represent an entirely novel insight.

*Reviewer #1:*

Lee et al. perform cytological analysis of GLP-1/Notch transcriptional targets *sygl-1* and *lst-1* to examine the relationship between transcriptional output and Notch signaling. They find that nascent, intron containing, *sygl-1* RNA is limited to the distal ~1-5 germ cell diameters, that cytoplasmic *sygl-1* mRNA is present in the distal ~1-10 cell diameters and that the presence of both nascent and cytoplasmic RNA is dependent on GLP-1/Notch activity. Lee et al. find that in a number of cells, within in the 1-5 cell diameters that shows evidence of GLP-1/Notch dependence, lack nascent intron containing *sygl-1*, which is interpreted as stochastic transcription. Nascent intron containing RNA from the *let-858* gene, which is not known to be spatially regulated, showed distal-to-proximal spatially unrestricted expression, which also can be interpreted as stochastic, as a number of nuclei lacked signal. The presence or absence of nascent *sygl-1* RNA was found to be independent of cell cycle stage. The spatial distribution of nascent and cytoplasmic *sygl-1* RNA is concordant but reduced in cell diameters, when analysis was performed in a mutant *glp-1* background that has partially reduced activity. Finally, using a temperature shift regimen, Lee et al. showed that the abundance of nascent and cytoplasmic *sygl-1* smFISH foci was a function of GLP-1/Notch activity, implying that as germ cells move farther away from the distal end, the GLP-1/Notch transcriptional output decreases.

This is a thorough and well controlled investigation of spatial control of GLP-1/Notch transcriptional targets, where the findings from Lee et al. are consistent with what was previously known and inferred from work from the Kimble, Schedl, Hubbard and Engebrecht groups.

The major assertion of the manuscript is that GLP-1/Notch signaling modulates transcriptional stochasticity of its targets, *sygl-1* and *lst-1*. The prevailing view from the field is that Pol II transcription of all genes in inherently stochastic [1], [2], [3] & [4], which is consistent with data presented for the non-spatially regulated gene *let-858*. In the case of *sygl-1*, which has an mRNA half-life of ~1hr, if there was continuous (non-stochastic) transcription, then there would be such high levels of cytoplasmic *sygl-1* mRNA that the stem cell control mechanism would likely become spatially unregulated.

Given the prevailing view, then essentially all mechanisms that upregulate transcription, like Notch or RTK signaling, must do so by modulating the frequency of transcriptional initiation/elongation, within the context of stochasticity. The extent of stochastic gene expression depends mostly on the mean level of that gene's expression [5], [6] & [7], which generates a linear relationship on a plot of log (ln) of variance of gene expression versus log (ln) mean of gene expression. A gene that shows specific regulation of stochastic expression would be found off the diagonal of the plot of log of variance of gene expression versus log mean of gene expression; *sygl-1* should show such an off-diagonal location based on the assertion of the authors. This reviewer appreciates that while an analysis to generate a plot of log variance of gene expression versus log mean gene expression is straightforward with genome wide approaches like RNA-seq, it would be very labor intensive to do the analysis cytologically, one gene at a time, as would be necessary for the system under study. Nevertheless, the information present in this manuscript is not sufficient to demonstrate that regulation of GLP-1/Notch targets is specifically through modulation of stochasticity.

1) Levsky, J.M. and R.H. Singer, Gene expression and the myth of the average cell. Trends in Cell Biology, 2003. 13(1): p. 4-6.

2) Li, G.-W. and X.S. Xie, Central dogma at the single-molecule level in living cells. Nature, 2011. 475(7356): p. 308-15.

3) Maheshri, N. and E.K. O'Shea, Living with noisy genes: how cells function reliably with inherent variability in gene expression. Annual Review of Biophysics and Biomolecular Structure, 2007. 36: p. 413-34.

4) Itzkovitz, S., et al., Single-molecule transcript counting of stem-cell markers in the mouse intestine. Nat Cell Biol. 14(1): p. 106-14.

5) Bar-Even, A., et al., Noise in protein expression scales with natural protein abundance. Nature Genetics, 2006. 38(6): p. 636-43.

6) Newman, J.R.S., et al., Single-cell proteomic analysis of *S. cerevisiae* reveals the architecture of biological noise. Nature, 2006. 441(7095): p. 840-6.

7) Vallania, F.L.M., et al., Origin and Consequences of the Relationship between Protein Mean and Variance. PloS One, 2014. 9(7): p. e102202.

Figure 3—figure supplement 1.

Comparisons made in panel D should not be grouped into 0-30μm, because as shown in Figure 3, there is as much as a two-fold difference in number of mRNAs based on distal – proximal position in the 0-30μm region. The comparisons should only be among cells at the same distance, in μm or cell diameters from the distal tip. Thus, while stochasticity is present, it is likely to be overestimated by the current approach.

*Reviewer #2:*

This paper examines the expression of Notch target genes in germline stem cells of the nematode *C. elegans*. It uses powerful fluorescent probes to detect what are likely single mRNAs, either nascent transcripts associated with active transcription sites (ATS) in the nucleus or mature spliced mRNAs in the cytoplasm. The microscopy and image analysis are carefully done, and the numerous controls leave little doubt that we are indeed looking at transcripts from Notch target genes. The authors also demonstrate convincingly that the *sygl-1* and *lst-1* transcriptional events are not coupled in a given cell, and that both ATS and cytoplasmic mRNA density are correlated with Notch signaling (as judged by a *glp-1* conditional mutation) and proximity to the distal tip (niche) cell.

This manuscript is technically impressive and it clarifies the relationship between transcription and steady-state mRNA levels in a developing animal tissue. Though one might imagine that a cell being stimulated by a signaling pathway would produce their various positively regulated target gene transcripts more or less constantly, it instead appears that at a given time only a minority of cells are actually doing so, at least at a level detectable using the imagining tools used here.

With the above overall enthusiastic evaluation in mind, I do have some concerns:

1) The authors present strong evidence to suggest that Notch targets are less transcribed as germ cells move away from the DTC niche cell. They note that this is correlated with cells being increasingly likely to lose contact with the finger-like processes of the DTC. This all makes sense given current models for germ cell differentiation. Another simple explanation (and not mutually exclusive with the authors') might be that DSL ligand proteins are also spatially graded along the processes of the DTC. Might the lovely myristoylated GFP probe the Kimble lab recently published (Byrd et al. 2014) be combined with a LAG-2 protein fusion like that described by Gengyo-Ando et al. (2006)?

2) In Figure 2, I am perplexed as to why there is not a bimodal distribution of DNA content, as would be expected if S-phase were relatively short compared to other parts of the cell cycle. Looking at the X axis, it is also a bit odd that the DAPI signal varies by five-fold, when the real difference cannot be greater than two-fold. This suggests that use of DAPI images for quantitation is very noisy. Why might that be? Permeability variation? Quenching by other fluorescent probes? Are there reasons to believe panel 2J might be more reliable? Some comment on this is probably in order.

3) I also have a statistical concern (see below).

*Reviewer #2 (Additional data files and statistical comments):*

The term "stochasticity" has been used to describe the observation that genetically identical cells in a common environment have variable gene expression at a given moment in time. Further, in eukaryotic cells a simple Poisson model (in which stochasticity is characterized by a constant probability of transcription over time) often fails to capture the dynamics of new transcription. Instead, a model of quiescence alternating with bursts of relatively frequent transcriptional initiation fits the data much better. Nevertheless, in a population of cells (or as a single cell is followed longitudinally over long periods of time) more predictable, uniform behaviors emerge. In this sense, transcriptional stochasticity is similar to how quantum-level phenomena lead to classical predictability in physics.

In this study of the *C. elegans* distal germ line, there is a predictable decline in cytoplasmic Notch target transcripts as cells move away from the DTC, yet even at the very tip ATS are only found in a few cells at any one time. Notch signaling could be seen as a parameter that increases the probability of a target gene ATS forming, but not as an "all on, all the time" situation. Provided the method used is not missing a large proportion of the actual ATS present, this is consistent with the above idea of stochasticity. As the authors state, "The transcriptional response to Notch signaling can vary from one receiving cell to another. This means that Notch signaling does not abolish the inherent stochasticity of transcription, but instead modulates that stochasticity."

The authors' interpretation is reasonable, but we could dig deeper to ask whether the dynamics are truly stochastic, and if so whether they are "simple" or "bursty," as has been found in other eukaryotic systems. The non-stochastic alternative goes like this:

Germ cells are all expressing Notch targets at the exact same rate, but they (and their two allelic copies of any given target gene) are out of phase with each other;

Most of a cycle of transcription is spent in the initiation phase;

When transcripts are elongated, they are rapidly spliced and exported to the cytoplasm;

Notch signaling functions to consistently shorten the initiation phase for all responding cells.

Because cells are asynchronous and the initiation phase is invisible, ATS would appear in a minority of cells, just as for a stochastic process. However, the canalized timing of ATS formation described above would produce a normally distributed range of ATS counts in replicated populations of cells at any given moment. In contrast, a one-state stochastic process would produce a Poisson-like distribution, while a multistate process would fit neither. I therefore encourage the authors to try some new analyses or simulations to see which of these alternative models fit their data best.

[Editors’ note: what now follows is the decision letter after the authors submitted for further consideration.]

Thank you for resubmitting your work entitled "*C. elegans* GLP-1/Notch activates transcription in a probability gradient across the germline stem cell pool" for further consideration at *eLife*. Your revised article has been favorably evaluated by Janet Rossant as the Senior editor and Reviewing Editor and two reviewers.

The manuscript has been improved but there are some remaining issues that the reviewers would like you to consider in your final submission. They are intended to improve your manuscript and we would appreciate your consideration of their inclusion.

*Reviewer 1:*

Lee et al. manipulated GLP-1/Notch activity through the use of a temperature sensitive reduction of function mutation. The authors also briefly described results with a gain of function GLP-1/Notch (Figure 1—figure supplement 2). A major question in the field is whether Notch gain of function mutants that, for example, contribute to cancer through disregulation of INTRA levels, have increased signaling activity or unregulated-ectopic signaling activity. From visual examination of Figure 1—figure supplement 2, it appears that the *glp-1* gain of function mutant has ATS signal distribution similar to wild type within the distal most 5 cell diameters, suggesting that this *glp-1* gain of function mutant, while displaying ectopic signaling does not have increased signaling activity in the distal most germ cells. Given that Lee et al. have n = 10 for this mutant they should be able to perform the same analysis described in the manuscript and provide important information as to whether the gain of function mutation has similar or increased signaling over wild type ligand dependent signaling in the distal most 5 cell diameters and more proximally.

*Reviewer 2:*

I believe the unresolved biological question we are all interested in is this: Why don't cells at a given distance from the DTC behave identically? One could even imagine a stripe of coordinated ATS-bearing cells N nuclei from the DTC, at least if the tissue were fixed at just the right time. After hundreds of such fixations we never see anything remotely like that. So, we can safely conclude that transcription of target gene copies is not coordinated in time. But, as I noted before, lack of coordination doesn't necessarily equate with randomness. It could be that each allele of a target gene in a field of cells experiencing the same level of Notch signal is following the exact same timetable for transcription, but because they are not coordinated to start with, heterogeneity is maintained. The simplest alternative is that the timetable itself is variable from allele to allele, but within some range given the amount of signaling. That is, there is a distribution of transcription cycle durations for any given level of Notch signaling. It's important to note that these alternatives exist even with a constant level of Notch signal. Were Notch signal strength to be increased, then in the first scenario the length of a uniform timetable is shortened, while in the latter there is a leftward shift in the distribution of timetable lengths, either of which would give predictably higher transcription population-wide.

It's my sense that the author's new language can accommodate either of the above scenarios, but if they agree with the alternatives above, then Figure 8 might be modified to reflect them. Because the *eLife* manuscript review interface doesn't allow uploading of files, I will submit my own attempt to do this to the editors via email in hopes that it be shared with the authors. I think the authors are leaning towards the second model, but we don't yet have the data to rule out the first.

---

## [Author Response]

[Editors’ note: the author responses to the first round of peer review follow.]

*[…] The reviewers and the editor all agree that the data presented are of very high quality and add to our understanding of Notch signaling in maintaining germ line stem cells in C. elegans. However, the claim of a novel linkage between cell signaling and stochastic transcription was felt not to be fully justified. The crux of your argument was interpreted as: 1) Not all cells are in the same state with respect to transcription, and therefore this means there is a stochastic process at work 2) Because there is nevertheless a clear impact of Notch-GLP-1 signaling on the distribution of transcription, "modulated stochasticity" is a key mechanism of signal transduction. However, mere non-homogeneity of transcription cannot on its own prove that stochastic forces are at work. When nascent transcripts are short-lived, and produced sparsely and asynchronously, many cells will lack them, yet the underlying process could actually be highly regular. In addition, there is now considerable existing evidence for the generality of stochasticity so that it is not clear that your findings represent an entirely novel insight.*

Major changes in the revised manuscript:

Stochasticity.

We thought that we were using the term “stochasticity” correctly, but based on the reviewers’ comments, we now consider our use of this term as incorrect and have removed it from the revised manuscript. As a result, we have reframed our conclusion to state that signaling does not orchestrate a uniform response in receiving cells, but that it instead activates transcription in a “probabilistic” manner at target genes and does so independently at each locus.

What is new:

The analysis of nascent transcripts at active transcription sites in cells responding to a canonical signaling pathway has not been before in any organism. I have presented our results at meetings and in seminars over the past year. These included a Cancer Research meeting on signaling and its control of stem cells, a Genetics Society meeting on model organisms and a Gordon Research Conference on Notch signaling. In all cases, the reaction to our findings was incredibly positive – ranging from “best talk of the meeting, it transformed my way of thinking about signaling”, to “we better start looking at our pathway this way as it is clearly the way to go” to “wow, we never would have thought that the germ cell response to Notch signaling would be graded like that”. When I presented the results at HHMI, Spyros was visibly jazzed and asked if I would be interested in helping with a meeting centered on Notch-regulated transcription. All in all, our results have gotten a very positive response from the relevant communities, consistent with them making an important advance forward.

We find that Notch signaling does not orchestrate a uniform response in receiving cells, but that it instead activates transcription in a “probabilistic” manner. Expectations varied considerably. Many people assumed that signaling would orchestrate a uniform response in receiving cells, and many others assumed that signaling would activate transcription in a “probabilistic” manner. The former group were largely cell or developmental biologists, many of whom work on developmental cell-cell signaling; the latter group were largely biochemists and molecular biologists. Regardless, this is the first study to provide quantitative data on the nuclear response to any canonical signaling pathway, and the answer is clear that Notch activates transcription in a “probabilistic” manner.

Discovery of a gradient in the probability of transcriptional activation is new. Gradients of course are not new, but a gradient of this type could not have been discovered without looking quantitatively at active transcription sites during signaling and no one had done that before.

Our analysis of different-strength NICDs demonstrates that a wild-type NICD drives a higher probability of transcriptional activation and larger number of nascent transcripts than a weaker mutant NICD. This may not be unexpected, but it has never been analyzed before and our results validate use of ATS as a new metric for Notch signaling and one that should be of general use. This metric opens a new approach for analysis of pathway mutants, including those with clinical effects, and also for engineering new pathway components, for example, to increase NICD strength or to act specifically on either probability or number of nascent transcripts.

What is unexpected:

Our finding that the shape of the ATS gradient is different from that of the mRNA gradient was not expected. My talk at the recent GRC on Notch signaling turned on a lot of light bulbs about the importance of assaying transcriptional activity in the nucleus as the most accurate assay for Notch-dependent transcriptional activation. Our manuscript now includes a figure highlighting this discrepancy (Figure 8) and argues explicitly that ATS provide a more direct and more accurate readout of canonical signaling.

Our finding that a steep ATS probability gradient can generate a near uniform and more extended field of mRNAs was not expected. Virtually all prior studies of canonical signaling analyzed mRNAs rather than ATS with many analyzing reporter mRNAs. A uniform field of mRNAs would have been interpreted as a uniform field of signaling. But now we find a graded signaling readout. This was a surprise!

We found a gradient in the probability of transcriptional activation that occurs across the *C. elegans* GSC pool, which is composed of developmentally equivalent cells. This was not expected. Cells in the pool are not only developmentally equivalent but they are surrounded by processes of the signaling distal tip cell. So the prevailing hypothesis was that the signaling response would be uniform. At the recent GSA meeting in Orlando, the many worm GSC experts who heard my talk were surprised and excited by these findings.

*Reviewer #1:*

*Lee et al perform cytological analysis of GLP-1/Notch transcriptional targets sygl-1 and lst-1 to examine the relationship between transcriptional output and Notch signaling. They find that nascent, intron containing, sygl-1 RNA is limited to the distal ~1-5 germ cell diameters, that cytoplasmic sygl-1 mRNA is present in the distal ~1-10 cell diameters and that the presence of both nascent and cytoplasmic RNA is dependent on GLP-1/Notch activity. Lee et al find that in a number of cells, within in the 1-5 cell diameters that shows evidence of GLP-1/Notch dependence, lack nascent intron containing sygl-1, which is interpreted as stochastic transcription. Nascent intron containing RNA from the let-858 gene, which is not known to be spatially regulated, showed distal-to-proximal spatially unrestricted expression, which also can be interpreted as stochastic, as a number of nuclei lacked signal. The presence or absence of nascent sygl-1 RNA was found to be independent of cell cycle stage. The spatial distribution of nascent and cytoplasmic sygl-1 RNA is concordant but reduced in cell diameters, when analysis was performed in a mutant glp-1 background that has partially reduced activity. Finally, using a temperature shift regimen, Lee et al showed that the abundance of nascent and cytoplasmic sygl-1 smFISH foci was a function of GLP-1/Notch activity, implying that as germ cells move farther away from the distal end, the GLP-1/Notch transcriptional output decreases.*

*This is a thorough and well controlled investigation of spatial control of GLP-1/Notch transcriptional targets, where the findings from Lee et al are consistent with what was previously known and inferred from work from the Kimble, Schedl, Hubbard and Engebrecht groups.*

Yes everything is consistent with what was previously known but our study takes regulation of this stem cell model to an entirely new level and makes a number of new insights, some of which could have been inferred and others that were not even imagined.

An example of a result that might been inferred is the effect of NICD strength on transcriptional activation. The GLP-1 q224 NICD was postulated to be weaker because of its mutant phenotype, and our study confirms that idea in nematodes. Prior to this work, NICD strength could not be quantitated at a molecular level in an intact organism. Our findings also validate use of ATS quantitation as a metric, as described in the “what’s new” section above.

An example of a result that had not even been proposed is the gradient in ATS probability across the stem cell pool. Prior to this work, a gradient in the probability of firing an ATS had not been seen in any developing tissue and it was certainly not proposed for this one. See “what’s new” and “what’s unexpected” sections above.

*The major assertion of the manuscript is that GLP-1/Notch signaling modulates transcriptional stochasticity of its targets, sygl-1 and lst-1. The prevailing view from the field is that Pol II transcription of all genes in inherently stochastic [1], [2], [3] & [4], which is consistent with data presented for the non-spatially regulated gene let-858. In the case of sygl-1, which has an mRNA half-life of ~1hr, if there was continuous (non-stochastic) transcription, then there would be such high levels of cytoplasmic sygl-1 mRNA that the stem cell control mechanism would likely become spatially unregulated.*

*Given the prevailing view, then essentially all mechanisms that upregulate transcription, like Notch or RTK signaling, must do so by modulating the frequency of transcriptional initiation/elongation, within the context of stochasticity. The extent of stochastic gene expression depends mostly on the mean level of that gene's expression [5], [6] & [7], which generates a linear relationship on a plot of log (ln) of variance of gene expression versus log (ln) mean of gene expression. A gene that shows specific regulation of stochastic expression would be found off the diagonal of the plot of log of variance of gene expression versus log mean of gene expression; sygl-1 should show such an off-diagonal location based on the assertion of the authors. This reviewer appreciates that while an analysis to generate a plot of log variance of gene expression versus log mean gene expression is straightforward with genome wide approaches like RNA-seq, it would be very labor intensive to do the analysis cytologically, one gene at a time, as would be necessary for the system under study. Nevertheless, the information present in this manuscript is not sufficient to demonstrate that regulation of GLP-1/Notch targets is specifically through modulation of stochasticity.*

*1) Levsky, J.M. and R.H. Singer, Gene expression and the myth of the average cell. Trends in Cell Biology, 2003. 13(1): p. 4-6.*

*2) Li, G.-W. and X.S. Xie, Central dogma at the single-molecule level in living cells. Nature, 2011. 475(7356): p. 308-15.*

*3) Maheshri, N. and E.K. O'Shea, Living with noisy genes: how cells function reliably with inherent variability in gene expression. Annual Review of Biophysics and Biomolecular Structure, 2007. 36: p. 413-34.*

*4) Itzkovitz, S., et al., Single-molecule transcript counting of stem-cell markers in the mouse intestine. Nat Cell Biol. 14(1): p. 106-14.*

*5) Bar-Even, A., et al., Noise in protein expression scales with natural protein abundance. Nature Genetics, 2006. 38(6): p. 636-43.*

*6) Newman, J.R.S., et al., Single-cell proteomic analysis of S. cerevisiae reveals the architecture of biological noise. Nature, 2006. 441(7095): p. 840-6.*

*7) Vallania, F.L.M., et al., Origin and Consequences of the Relationship between Protein Mean and Variance. PloS One, 2014. 9(7): p. e102202.*

In response to this comment, we have reframed our major conclusion. We now conclude that signaling does not orchestrate a uniform response in receiving cells, but that it instead activates transcription in a “probabilistic” manner at individual target genes and does so independently at each locus. This reviewer mentions that a plot of log variance of gene expression versus log mean gene expression is a common way to test for stochasticity of gene expression. However, that analysis is done with RNA-Seq data and focuses on mRNAs, which are not applicable to our ATS analysis. Instead, we followed the suggestion of reviewer #2 to estimate stochasticity of *sygl-1* ATS by comparing their spatial pattern to a Poisson random simulation (see below).

Figure 3—figure supplement 1.

*Comparisons made in panel D should not be grouped into 0-30μm, because as shown in Figure 3, there is as much as a two-fold difference in number of mRNAs based on distal - proximal position in the 0-30μm region. The comparisons should only be among cells at the same distance, in um or cell diameters from the distal tip. Thus, while stochasticity is present, it is likely to be overestimated by the current approach.*

We do not understand this comment because we never argued for stochasticity of mRNA abundance. We also note that the revised manuscript no longer focuses on stochasticity.

*Reviewer #2:*

*[…] 1) The authors present strong evidence to suggest that Notch targets are less transcribed as germ cells move away from the DTC niche cell. They note that this is correlated with cells being increasingly likely to lose contact with the finger-like processes of the DTC. This all makes sense given current models for germ cell differentiation. Another simple explanation (and not mutually exclusive with the authors') might be that DSL ligand proteins are also spatially graded along the processes of the DTC. Might the lovely myristoylated GFP probe the Kimble lab recently published (Byrd et al. 2014) be combined with a LAG-2 protein fusion like that described by Gengyo-Ando et al. (2006)?*

Yes, we find an ATS gradient, but in fact we do not find the correlation with loss of DTC contact that the reviewer suggests. Just the opposite. The DTC or its processes contact virtually all cells with ATS as well as the lion’s share of cells lacking ATS (see Figure 3 response to reviewer #1). In unpublished work, we have found that a LAG-2::GFP fusion protein is visible in all DTC processes (both short intercalating as well as long external processes), but this is not terribly informative – it tells us where a transgenic DSL ligand is expressed, but not where that ligand is triggering a response. The transgenic LAG-2::GFP rescues a *lag-2* null mutant, but its biological phenotype is not fully wild-type so we prefer to CRISPR tag endogenous LAG-2 and APX-1, which is on our list of things to do, but not yet accomplished. More importantly, *sygl-1* and *lst-1* ATS provide an unprecedently accurate readout of where signaling has achieved a transcriptional response and this is the focus of our paper.

*2) In Figure 2, I am perplexed as to why there is not a bimodal distribution of DNA content, as would be expected if S-phase were relatively short compared to other parts of the cell cycle. Looking at the X axis, it is also a bit odd that the DAPI signal varies by five-fold, when the real difference cannot be greater than two-fold. This suggests that use of DAPI images for quantitation is very noisy. Why might that be? Permeability variation? Quenching by other fluorescent probes? Are there reasons to believe panel 2J might be more reliable? Some comment on this is probably in order.*

S-phase is not short relative to other phases of the cell cycle. Instead, G1 is very, very short and S-phase consumes about ~70% of the length of the cell cycle, as we published in an *eLife* paper last year (Seidel and Kimble, 2015). The revised manuscript now includes the lengths of the various cell cycle phases to address this misconception. As for the DAPI signal variability, the difference between low and high DAPI intensity value is ~2.6-fold after we remove outliers (data points more than 1.5 interquartile range); this number is closer to what one might expect. Possible sources of error that might generate outliers include variation in DAPI staining from gonad to gonad and capture of nuclear images every 0.3 µm on the z-axis. Moreover, we used nuclear size as another metric to support the DAPI measurement. All in all, our data strongly support our conclusion that activation of *sygl-1* ATS does not correlate with any phase of the cell cycle. We note that this result was also not expected, at least by some workers in Notch signaling, but this surprise is sufficiently minor that we did not include it in the “not expected” results above.

*3) I also have a statistical concern (see below).*

*Reviewer #2 (Additional data files and statistical comments):*

*The term "stochasticity" has been used to describe the observation that genetically identical cells in a common environment have variable gene expression at a given moment in time. Further, in eukaryotic cells a simple Poisson model (in which stochasticity is characterized by a constant probability of transcription over time) often fails to capture the dynamics of new transcription. Instead, a model of quiescence alternating with bursts of relatively frequent transcriptional initiation fits the data much better. Nevertheless, in a population of cells (or as a single cell is followed longitudinally over long periods of time) more predictable, uniform behaviors emerge. In this sense, transcriptional stochasticity is similar to how quantum-level phenomena lead to classical predictability in physics.*

*In this study of the C. elegans distal germ line, there is a predictable decline in cytoplasmic Notch target transcripts as cells move away from the DTC, yet even at the very tip ATS are only found in a few cells at any one time. Notch signaling could be seen as a parameter that increases the probability of a target gene ATS forming, but not as an "all on, all the time" situation. Provided the method used is not missing a large proportion of the actual ATS present, this is consistent with the above idea of stochasticity. As the authors state, "The transcriptional response to Notch signaling can vary from one receiving cell to another. This means that Notch signaling does not abolish the inherent stochasticity of transcription, but instead modulates that stochasticity."*

*The authors' interpretation is reasonable, but we could dig deeper to ask whether the dynamics are truly stochastic, and if so whether they are "simple" or "bursty," as has been found in other eukaryotic systems. The non-stochastic alternative goes like this:*

*Germ cells are all expressing Notch targets at the exact same rate, but they (and their two allelic copies of any given target gene) are out of phase with each other;*

*Most of a cycle of transcription is spent in the initiation phase;*

*When transcripts are elongated, they are rapidly spliced and exported to the cytoplasm;*

*Notch signaling functions to consistently shorten the initiation phase for all responding cells.*

*Because cells are asynchronous and the initiation phase is invisible, ATS would appear in a minority of cells, just as for a stochastic process. However, the canalized timing of ATS formation described above would produce a normally distributed range of ATS counts in replicated populations of cells at any given moment. In contrast, a one-state stochastic process would produce a Poisson-like distribution, while a multistate process would fit neither. I therefore encourage the authors to try some new analyses or simulations to see which of these alternative models fit their data best.*

In the originally submitted manuscript we used the term “stochasticity” to describe the independent Notch-dependent transcription of its target genes at chromosomal loci. We appreciate suggestions to “dig deeper” into whether the ATS dynamics are truly stochastic, but suggest that this issue is tangential to our reframed conclusion that Notch signaling does not orchestrate a uniform response in receiving cells, but that it instead activates transcription in a “probabilistic” manner at individual target genes and does so independently at each locus. The data backing up this conclusion are strong: lack of correlation between intensities of ATS at loci within single nuclei for both *sygl-1* (Figure 2) and *lst-1* (Figure 4), lack of correlation between numbers of *sygl-1* and *lst-1* ATS in the same nucleus (Figure 4) and lack of correlation of the summed ATS intensities of *sygl-1* and *lst-1* in the same nucleus (Figure 4).

We appreciate reviewer #2’s thoughtful suggestions about testing stochasticity of ATS timing but respectfully suggest that these tests are premature. Although we can do simulations, the experimental data for comparison do not yet exist. We are generating constructs to attempt live imaging of *sygl-1* and *lst-1* ATS and hope to have that data at some point in the future. However, these experiments rely on introduction of many reiterated copies of MS2 binding sites, and such reiterated sequences are often silenced in *C. elegans* germ cells. Thus, we will try but do not know if and when it will work. Once we have overcome this hurdle and have data for frequency and/or duration of transcriptional pulses, we can do the requested simulations and test this question.

Nonetheless, we have done two things in an attempt to address Reviewer 2’s concerns. First, we added Figure 8, which illustrates models for ATS timing possibilities even though testing them lies in the future. Second, we did simulations based on our current data to demonstrate that ATS are not generated stochastically. Given the steep ATS gradient, this is no surprise and we do not suggest that these simulations be incorporated into the paper. Because the reviewers might be interested, we have included them below.

Author response image 1.Tests to ask if positions of sygl-1 ATS is stochastic among cells in the GSC pool.A, B) We recorded positions of *sygl-1* ATS-positive and -negative cells within the region harboring ATS (0-30 µm from distal end) for each gonad examined (n=60 gonads) and then used a Poisson distribution to position ATS-positive cells randomly. We conducted this random simulation 100 times for each gonad to generate a dataset of “Complete Spatial Randomness” (CSR) for ATS-positive cells, which establishes a baseline for true stochasticity. (**A**) We compared the random CSR simulations with actual ATS data using 3-D Ripley’s H function, a widely used method for analyzing spatial patterns that assesses degrees of clustering. This analysis shows that the CSR clustering was 0, which differs significantly from the clustering seen with the actual ATS data. **p-value < 0.01 by t-test. n = 78 gonads. (**B**) We compared distance of ATS-positive cells from the distal end for the random CSR simulations and the actual data, and find that the CSR data differ significantly from the actual data. The actual ATS-positive cells are generally closer to the distal end than CSR data, which supports the non-random ATS clustering seen in A. *****p < 0.00001 by t-test. n = 2759 nuclei from 78 gonads.**DOI:**
http://dx.doi.org/10.7554/eLife.18370.017

[Editors’ note: the author responses to the re-review follow.]

*[…] The manuscript has been improved but there are some remaining issues that the reviewers would like you to consider in your final submission. They are intended to improve your manuscript and we would appreciate your consideration of their inclusion.*

*Reviewer 1:*

*Lee et al. manipulated GLP-1/Notch activity through the use of a temperature sensitive reduction of function mutation. The authors also briefly described results with a gain of function GLP-1/Notch (Figure 1—figure supplement 2). A major question in the field is whether Notch gain of function mutants that, for example, contribute to cancer through disregulation of INTRA levels, have increased signaling activity or unregulated-ectopic signaling activity. From visual examination of Figure 1—figure supplement 2, it appears that the glp-1 gain of function mutant has ATS signal distribution similar to wild type within the distal most 5 cell diameters, suggesting that this glp-1 gain of function mutant, while displaying ectopic signaling does not have increased signaling activity in the distal most germ cells. Given that Lee et al. have n = 10 for this mutant they should be able to perform the same analysis described in the manuscript and provide important information as to whether the gain of function mutation has similar or increased signaling over wild type ligand dependent signaling in the distal most 5 cell diameters and more proximally.*

We have now done this analysis and included it as a new supplementary figure: Figure 3—figure supplement 1. Our results show that the percentage of cells with *sygl-1* ATS is uniform in the strain with the unregulated receptor (Figure 3—figure supplement 1) and that the intensity is also about the same as the intensity of ATS in wild-type (Figure 3—figure supplement 1). Because this *glp-1(gf)* analysis had to be done in L4 gonads for technical reasons, we also include in this supplementary figure two additional panels to document the response in wild-type L4 germ cells, which is very much like wild-type adult germ cells (Figure 3—figure supplement 1).

We also revised the text to describe this new data (–subsection “Gradient of Notch-dependent *sygl-1* transcriptional activation”, last paragraph) and include a legend for the Figure 3—figure supplement 1).

*Reviewer 2:*

*I believe the unresolved biological question we are all interested in is this: Why don't cells at a given distance from the DTC behave identically? One could even imagine a stripe of coordinated ATS-bearing cells N nuclei from the DTC, at least if the tissue were fixed at just the right time. After hundreds of such fixations we never see anything remotely like that. So, we can safely conclude that transcription of target gene copies is not coordinated in time. But, as I noted before, lack of coordination doesn't necessarily equate with randomness. It could be that each allele of a target gene in a field of cells experiencing the same level of Notch signal is following the exact same timetable for transcription, but because they are not coordinated to start with, heterogeneity is maintained. The simplest alternative is that the timetable itself is variable from allele to allele, but within some range given the amount of signaling. That is, there is a distribution of transcription cycle durations for any given level of Notch signaling. It's important to note that these alternatives exist even with a constant level of Notch signal. Were Notch signal strength to be increased, then in the first scenario the length of a uniform timetable is shortened, while in the latter there is a leftward shift in the distribution of timetable lengths, either of which would give predictably higher transcription population-wide.*

*It's my sense that the author's new language can accommodate either of the above scenarios, but if they agree with the alternatives above, then Figure 8 might be modified to reflect them. Because the eLife manuscript review interface doesn't allow uploading of files, I will submit my own attempt to do this to the editors via email in hopes that it be shared with the authors. I think the authors are leaning towards the second model, but we don't yet have the data to rule out the first.*

We agree that either of these two scenarios is possible but have no real reason to favor one over the other. To address this suggestion, we therefore added the text as follows:

“We do not yet know if Notch regulates frequency or duration of these pulses, or if timing of Notch-dependent transcriptional activation follows a set rhythm once initiated. Answers to these questions must await live imaging of nascent RNAs.”

We also considered the idea of making this point in a revised Figure 8, but decided that this addition would complicate an already speculative figure with further speculation.